# *Bacillus megaterium* HgT21: a Promising Metal Multiresistant Plant Growth-Promoting Bacteria for Soil Biorestoration

Jesús Guzmán-Moreno,[a] ⓘ Luis Fernando García-Ortega,[b] Lilia Torres-Saucedo,[a] Paulina Rivas-Noriega,[a] Rosa María Ramírez-Santoyo,[a] Lenin Sánchez-Calderón,[c] Iliana Noemi Quiroz-Serrano,[a] ⓘ Luz Elena Vidales-Rodríguez[a]

[a]Laboratorio de Biología de Bacterias y Hongos Filamentosos, Unidad Académica de Ciencias Biológicas, Universidad Autónoma de Zacatecas, Zacatecas, Zacatecas, Mexico
[b]Departamento de Ingeniería Genética, Centro de Investigación y de Estudios Avanzados del Instituto Politécnico Nacional (CINVESTAV), Irapuato, Guanajuato, Mexico
[c]Laboratorio de Genómica Evolutiva, Unidad Académica de Ciencias Biológicas, Universidad Autónoma de Zacatecas, Zacatecas, Zacatecas, Mexico

**ABSTRACT** The environmental deterioration produced by heavy metals derived from anthropogenic activities has gradually increased. The worldwide dissemination of toxic metals in crop soils represents a threat for sustainability and biosafety in agriculture and requires strategies for the recovery of metal-polluted crop soils. The biorestoration of metal-polluted soils using technologies that combine plants and microorganisms has gained attention in recent decades due to the beneficial and synergistic effects produced by its biotic interactions. In this context, native and heavy metal-resistant plant growth-promoting bacteria (PGPB) play a crucial role in the development of strategies for sustainable biorestoration of metal-contaminated soils. In this study, we present a genomic analysis and characterization of the rhizospheric bacterium *Bacillus megaterium* HgT21 isolated from metal-polluted soil from Zacatecas, Mexico. The results reveal that this autochthonous bacterium contains an important set of genes related to a variety of operons associated with mercury, arsenic, copper, cobalt, cadmium, zinc and aluminum resistance. Additionally, halotolerance-, beta-lactam resistance-, phosphate solubilization-, and plant growth-promotion-related genes were identified. The analysis of resistance to metal ions revealed resistance to mercury ($Hg^{II+}$), arsenate $[AsO_4]^{3-}$, cobalt ($Co^{2+}$), zinc ($Zn^{2+}$), and copper ($Cu^{2+}$). Moreover, the ability of the HgT21 strain to produce indole acetic acid (a phytohormone) and promote the growth of *Arabidopsis thaliana* seedlings *in vitro* was also demonstrated. The genotype and phenotype of *Bacillus megaterium* HgT21 reveal its potential to be used as a model of both plant growth-promoting and metal multiresistant bacteria.

**IMPORTANCE** Metal-polluted environments are natural sources of a wide variety of PGPB adapted to cope with toxic metal concentrations. In this work, the bacterial strain *Bacillus megaterium* HgT21 was isolated from metal-contaminated soil and is proposed as a model for the study of metal multiresistance in spore-forming Gram-positive bacteria due to the presence of a variety of metal resistance-associated genes similar to those encountered in the metal multiresistant Gram-negative *Cupriavidus metallidurans* CH34. The ability of *B. megaterium* HgT21 to promote the growth of plants also makes it suitable for the study of plant-bacteria interactions in metal-polluted environments, which is key for the development of techniques for the biorestoration of metal-contaminated soils used for agriculture.

**KEYWORDS** *Bacillus megaterium*, metal resistance, plant growth-promoting bacteria, metal contamination

Address correspondence to Luz Elena Vidales-Rodríguez, luzelenavr@uaz.edu.mx.

The authors declare no conflict of interest.

H eavy metals are present in the environment due to a variety of natural processes, such as volcanic activity, weathering of rocks, and windblown dust particles (1). However, the production of heavy metal wastes derived from mining and other anthropogenic

activities has significantly increased the concentration and bioavailability of toxic metals in ecosystems (1, 2). Some metals (i.e., Zn, Fe, Co, Ni, Cu, Mg, and Mn) are essential at low concentrations for the metabolic activities of living organisms; however, the homeostasis of these metals must be tightly regulated by cells to avoid their toxic effects (3). On the other hand, heavy metals without known biological functions (i.e., Hg, Pb, Ag, Cr, Cd) are highly toxic even at low concentrations for most living organisms, particularly for microbial and plant communities in soils, and constant exposure to toxic metals alters their composition, growth, activity, and genetic variability (4, 5).

Despite the toxicity caused by metals, a variety of bacteria that have been constantly exposed to toxic metal ions have tolerance/resistance mechanisms that enable them to survive and colonize hostile environments (6). Bacterial resistance/tolerance to metal ions generally involves extracellular complexation by sorption on the cell surface, intracellular metal accumulation and complexation through intracellular binding to metallothioneins and chelation by siderophores, the intracellular redox of metal ions by enzymatic activity, and cellular exclusion by efflux systems (5, 7).

Some metal-resistant bacteria belong to well-described bacterial genera known as plant growth-promoting bacteria (PGPB), which in addition to modifying the toxic effects that metal ions produce in plants (8–10), also promote plant growth in contaminated environments through a variety of mechanisms, including nutrient mobilization, degradation of organic substances, phosphate solubilization, nitrogen fixation, antibiotic production, iron sequestration, and the production of vitamins and phytohormones such as gibberellins and auxins (11–13).

The *Bacillus* genus of bacteria includes a wide variety of species recognized as PGPB (14, 15); among these, *Bacillus megaterium* (*Priestia megaterium*) is one of the most studied and is a common inhabitant of soil that is frequently associated with metal-contaminated environments (16, 17). In *B. megaterium*, the metal tolerance associated with biosorption and siderophore production has been described for metals such as Ni (18), Cd (10, 19, 20), Pb (19, 20), Cu, and Zn (17). However, except for the mercury resistance operon (*mer*), which has been fully studied (21, 22), few studies of metal resistance determinants and mechanisms in *B. megaterium* have been reported. These studies include a description of resistance genes for Cd (17, 23, 24), Cu (25), Pb (17), and Ni (26). This work was focused on the characterization of the Hg-tolerant bacterial strain HgT21, isolated from the rhizosphere of a metal-contaminated soil. Genome analysis was carried out to determine its potential for use as a plant growth promoter in metal-contaminated soils on which extreme living conditions are prevalent. Phylogeny and genome analyses and *in vitro* assays indicate that the isolate HgT21 identified as *B. megaterium* contains a battery of genes and operons that confer resistance to a variety of heavy metals(oids) and several traits related to halotolerance and plant growth promotion. The results reveal that *B. megaterium* HgT21 can be used as a model for the study of metal multiresistance in Gram-positive bacteria, with application as a PGPB for phytoimmobilization and restoration of metal-contaminated sites.

## RESULTS

**Isolation of the mercury-tolerant HgT21 bacterial strain.** The $Hg^{II+}$ tolerance of the HgT21 strain was determined using the MIC for mercuric chloride ($HgCl_2$) in both agar and liquid LB medium. The MIC value in solid medium was 975 $\mu$M, whereas in liquid medium, bacterial growth was fully inhibited at 75 $\mu$M (Fig. 1).

**Genome sequencing, *de novo* assembly, and annotation.** The whole-genome sequence of strain HgT21 was obtained using the Illumina platform with MiSeq Illumina paired-end technology with 300-bp reads. Our assembly consisted of 26 contigs spanning $\sim$5.5 Mbp with an average GC content of 37.6% and an $N_{50}$ size of 1,202,273 bp ($\sim$1.2 Mb). Genome annotation by Rapid Annotation using Subsystem Technology (RAST) predicted 5,741 protein coding DNA sequences (CDSs) assigned to 351 subsystems. The subsystems feature genes including bacteriocins and antibacterial peptides ($n = 9$), membrane transport systems ($n = 97$), auxin biosynthesis ($n = 5$), nitrogen metabolism ($n = 25$), metabolism of aromatic compounds ($n = 16$), iron acquisition and metabolism ($n = 65$), phosphorus

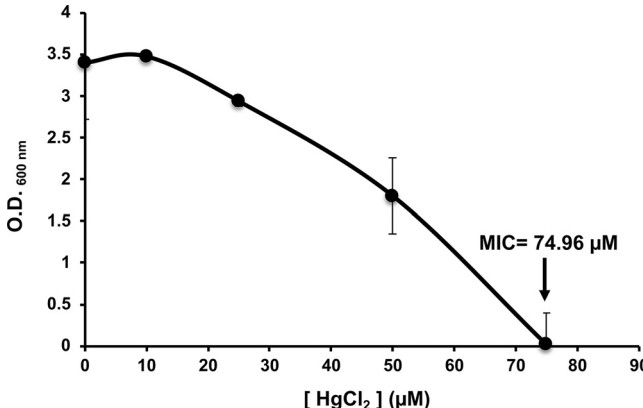

**FIG 1** Hg tolerance of the HgT21 strain in liquid medium. HgT21 bacterial cells were aerobically grown in LB liquid medium supplemented with increasing concentrations of $HgCl_2$. The optical density at 600 nm ($O.D._{600nm}$) was measured, and the concentration at which bacterial growth was fully inhibited by mercury ions (MIC) was determined (black arrow).

metabolism ($n = 45$), resistance to antibiotic and toxic compounds ($n = 87$), and stress response ($n = 163$). Additionally, the annotation process detected a total of 34 rRNA genes, 114 tRNAs, 117 ribozymes, and 6 prophage regions, 3 of which were identified as active prophages (Fig. 2). No plasmid was identified when analyzed using PLACNETw and Plasmid Finder.

**Morphological and biochemical characterization of the HgT21 strain.** The morphology and biochemical characteristics, such as the bacillus shape, spore formation, positive Gram staining, acid production from glucose, and the use of mannitol and citrate as carbon sources, as well as the presence of catalase, amylase, and oxidase enzymatic activities (Table 1), identified the HgT21 strain as a member of the *Bacillus* genus based on characteristics reported for this genus (27, 28). Additional enzymatic activities and biochemical characteristics related to carbohydrate, polyalcohol, and xenobiotic metabolism are shown in Table 1.

**Identification and phylogenetic analysis of HgT21.** The HgT21 strain was identified as *Bacillus megaterium* by whole-genome multilocus sequence typing (wgMLST) analysis. A rooted species tree was inferred for HgT21 and 28 closely related species (see Fig. S1 in the supplemental material) using 8,098 orthogroups. Phylogenetic analysis revealed the close evolutionary relationship of HgT21 with *B. megaterium* DSM-319 (Fig. S2). In general, the close evolutionary relationship between *B. megaterium* and *Bacillus aryabhattai* strains exhibited in the tree is in agreement with previous studies (29, 30). The tree topology groups the strains of *B. aryabhattai* and *B. megaterium* into two main clades, with *B. megaterium* WSH-002 as a common ancestor of these clades.

**Core and pangenome of the *B. megaterium*-*B. aryabhattai* clade.** To determine the genomic plasticity and global gene reservoir of the clade *B. megaterium*-*B. aryabhattai*, a pangenome analysis was performed based on the annotated protein sequences of the 29 strains (Fig. S3). The 157,299 protein sequences present across all genomes were clustered into 15,974 orthogroups, representing the pangenome. Among them, 1,633 orthogroups (10.22%) were conserved in all 29 genomes, representing the core genome of all species analyzed (all-core), and 1,229 orthogroups (7.69%) of the core genome retained only one copy in every strain. Moreover, we identified 3,341 orthogroups (20.91%) conserved in the clade *B. megaterium*-*B. aryabhattai* (excluding *B. aryabhattai* PHB10) (MA-core), 3,007 of which (90%) were orthogroups of single-copy genes. On the other hand, 1,954 (12.23%) orthogroups were conserved among the outgroup clade (93.5% single copy) (out-core). To determine the relationship of core genes with some evolutionary features, we classified these genes into different functional categories using Clusters of Orthologous Genes (COG) annotation. We assigned 1,621, 1,309, and 1,166 functions for 80.36%, 66.83%, and 71.4% of proteins in the MA, out, and all-gene cores, respectively. Proteins involved in general function dominated the three cores analyzed (Fig. S4). Some of the most abundant proteins in this class were related to predicted

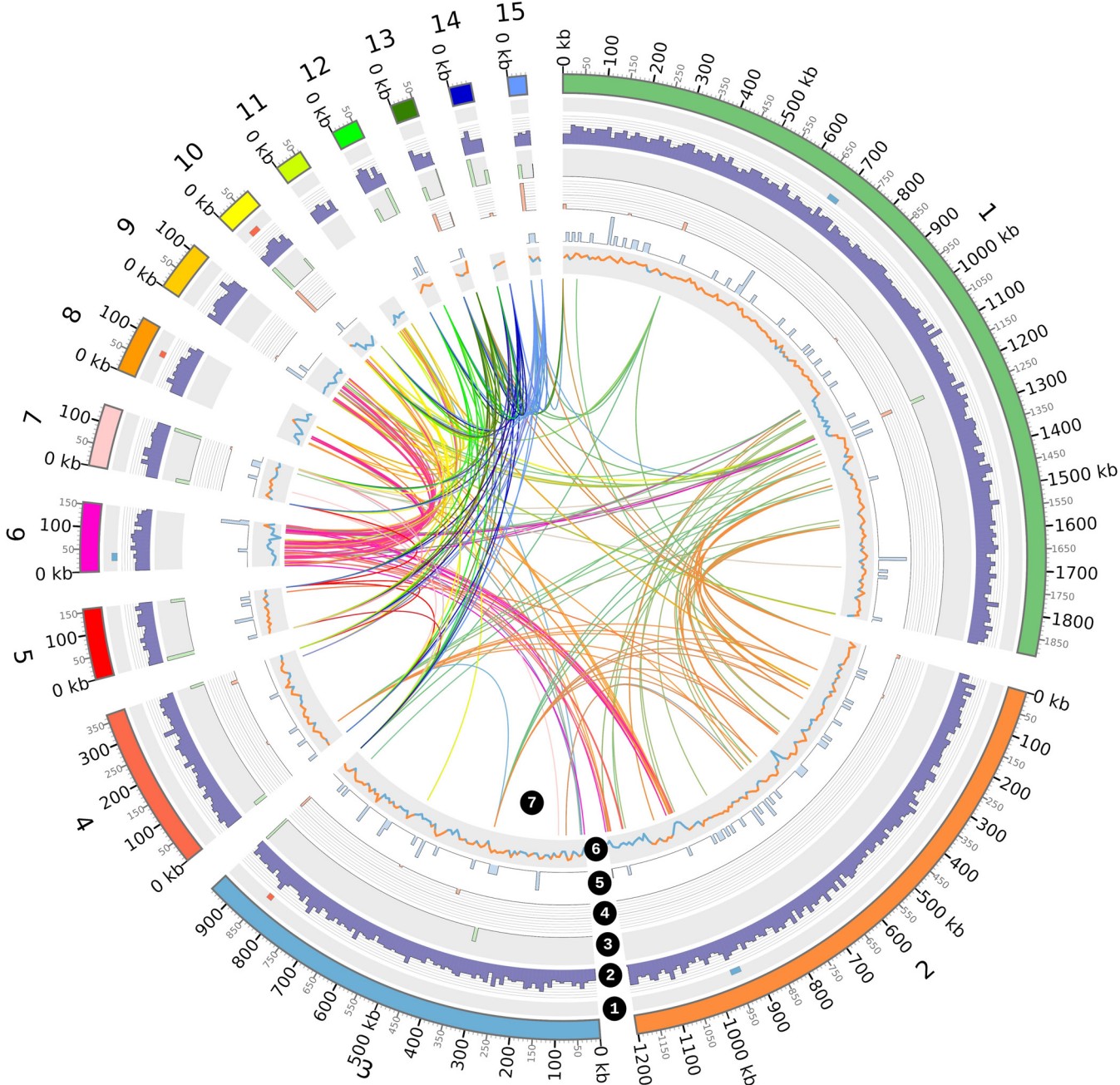

**FIG 2** *Bacillus megaterium* HgT21 draft genome landscape. From the outside to inside, the rings correspond to the following: (1) the tile plot depicts the detected prophage regions in the genome, red tiles represent active prophages, and blue tiles indicate inactive or ambiguous regions; (2) gene density; (3) rRNAs; (4) tRNAs; (5) ncRNAs; and (6) GC content (orange, above average; light blue, below average). All the statistics are based on 10-kb nonoverlapping bins. Links between scaffolds represent colinear blocks (7). Only scaffolds with a length of >20 Kb were plotted.

hydrolases of the haloacid dehydrogenase (HAD) superfamily, Zn-dependent hydrolases, including glyoxylases, and predicted permease, a member of the PurR regulon. Through comparison between COG clusters, we found that transcription, secondary metabolite biosynthesis, transport, and catabolism and signal transduction mechanisms were more abundant in the MA core than in the other cores. Translation, ribosomal structure and biogenesis, replication, recombination and repair, and nucleotide transport and metabolism were the least abundant (Fig. S4).

The distribution of accessory genes also varied among strains as well as clades (Fig. 3). A total of 7,333 (45.9%) orthogroups were singletons present in only one strain, which was reduced to 490 (3.06%) when considering only the *B. megaterium-B. aryabhattai* clade. A

**TABLE 1** Morphology and biochemical characteristics of the HgT21 strain

| Characteristics | Results[a] |
|---|---|
| Morphology and general characteristics | |
| Colony | Round, regular edge, beige, smooth, convex, 4–6 mm diam |
| Cell morphology and Gram stain | Rod-like long bacilli, Gram-positive |
| Spore formation | + |
| Motility | + |
| $H_2S$ production | − |
| | |
| Biochemical characteristics | |
| Carbohydrate metabolism | |
| Acid from glucose | + |
| Voges-Proskauer reaction | + |
| Citrate | + |
| D-raffinose | + |
| D-xylose | + |
| D-galactose | + |
| D-ribose | + |
| L-lactato alcalinization | − |
| Lactose | + |
| Saccharose/sucrose | + |
| D-maltose | + |
| D-trehalose | + |
| D-mannose | − |
| Cyclodextrine | − |
| Pullulane | − |
| Polyalcohol metabolism | |
| D-sorbitol | + |
| Mannitol | + |
| Xenobiotic metabolism | |
| D-amygdalin | − |
| Salicin | + |
| | |
| Enzymatic activity | |
| Amylase | + |
| Catalase | + |
| Casein Hydrolysis | + |
| Lecithinase | − |
| Urease | − |
| Phosphatase | − |
| Cytochrome oxidase | + |
| Lysine descarboxylase | + |
| Ornithine descarboxylase | + |
| Tryptophanase | − |
| Arginine dihydrolase 1 | + |
| Arginine dihydrolase 2 | − |
| Ala-phe-pro arylamidase | − |
| L-aspartate arylamidase | − |
| Leucine arylamidase | − |
| L-proline arylamidase | − |
| L-pyrrolydonyl arylamidase | − |
| Alanine arylamidase | + |
| Tyrosine arylamidase | + |
| Beta galactosidase | + |
| Alpha galactosidase | + |
| Alpha glucosidase | + |
| Alpha-mannosidase | − |
| Beta glucuronidase | − |
| Beta galactopyranosidase resofurine | + |
| *N*-acetyl-D-glucosaminidase | + |
| Metyl-B-D-glucopyranosidase | − |
| Phosphatidylinositol phospholipase C | − |

[a]+, positive; −, negative.

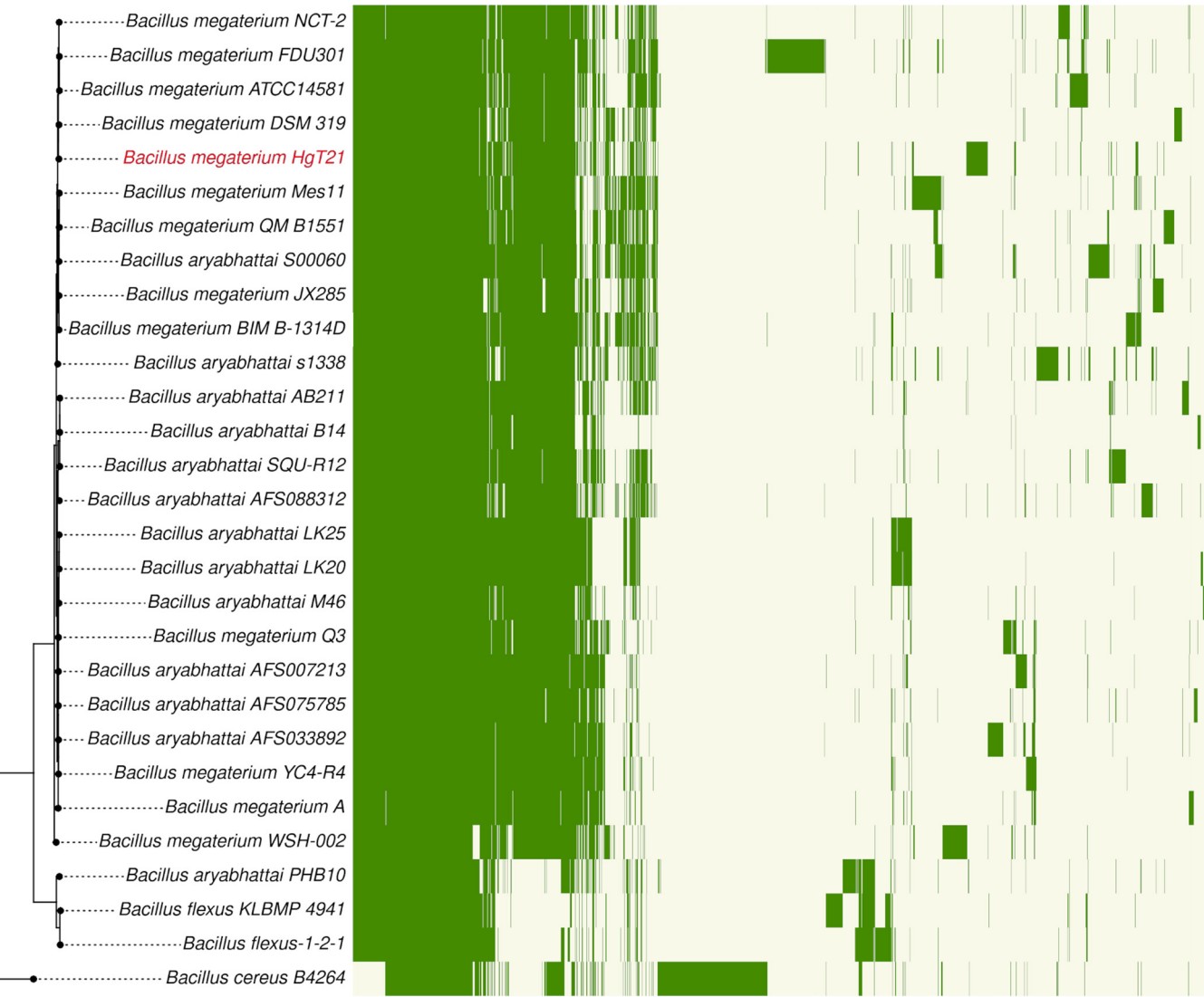

**FIG 3** Phylogenetic tree of the species closely related to HgT21 against the gene presence/absence matrix. The heatmap on the right shows the presence (dark green) or absence (light yellow) of all 15,974 orthogroups. Each row in the matrix corresponds to a branch on the tree (i.e., one species), and each column represents an orthogroup.

total of 332 strain-specific genes were identified in *B. megaterium* HgT21, including 230 annotated hypothetical proteins (Table S2).

**Relevant metal resistance operons in HgT21.** *B. megaterium* HgT21 genome analysis revealed the presence of several genes involved in heavy metal resistance. These genes are associated with resistance to arsenic, copper, mercury, tellurium, zinc, cadmium, and cobalt and are grouped as operons (Fig. 4). Two putative arsenic resistance operons were found (contigs 2 and 8; Fig. 4). The genes *arsC*, *arsB*, and *arsR* and an extra copy of *arsR* were found at contig 2, whereas *arsA*, *arsD*, *arsC*, *arsB*, *arsR*, and *arsB-acr3* and an extra copy of *arsR* were found at contig 8. Additionally, two putative identical copper resistance-like operons (*csoR-copZ-copA*) were found (contigs 3 and 8; Fig. 4), as well as a putative mercury resistance operon (*merRETPA*) and the *merB* gene (contig 8; Fig. 4). A putative tellurium resistance operon that includes *telA*, *yceG*, and three tandem copies of *telD* was found (contig 3; Fig. 4). Finally, three genes associated with Zn-Cd-Co resistance (*zntA*, *arsR*, and *czcD*) were identified. Overall, these genes and operons showed gene arrangements similar to those reported in bacteria, with some differences in the cases of cadmium, arsenic, and tellurium (22, 31–33).

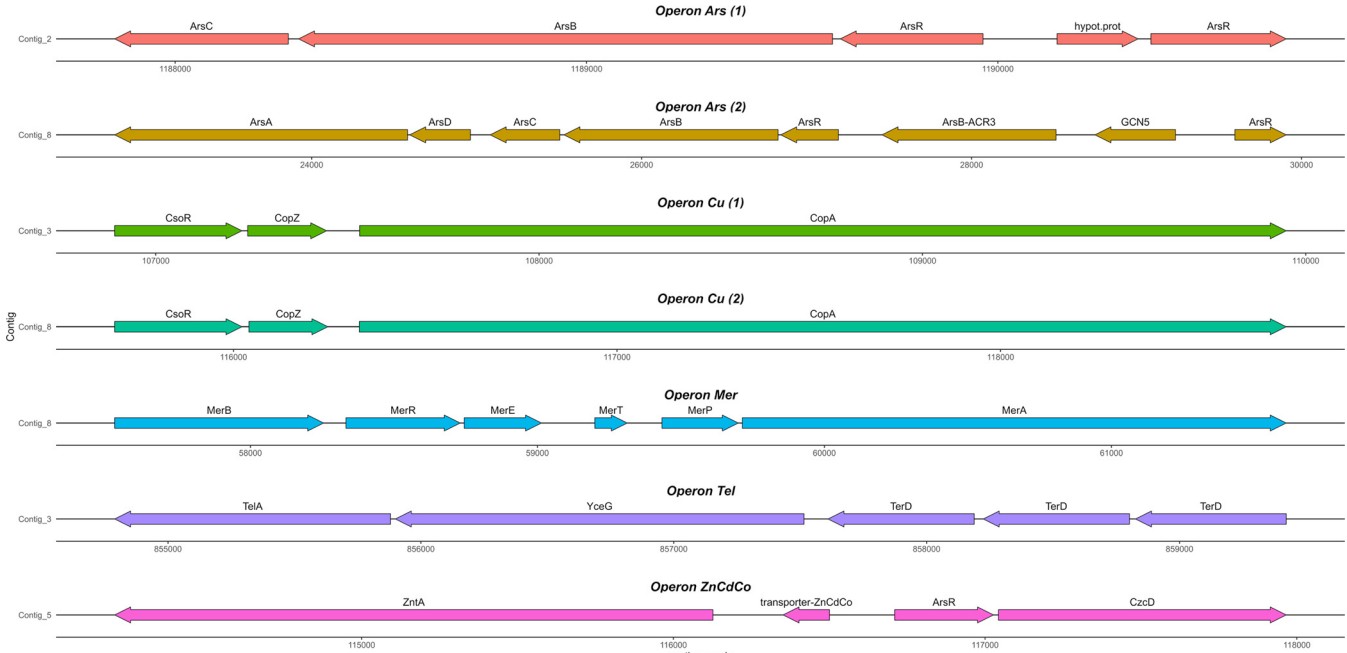

**FIG 4** Metal operons in *B. megaterium* HgT21. The locations, orientations, and products of putative operons involved in heavy metal resistance determinants. Annotated genes attributed to arsenic 1, arsenic 2, copper 1, copper 2, mercury, tellurium, and zinc-cadmium-cobalt are displayed in coral, brown, green, cyan, blue, purple and pink, respectively.

**Metal resistance of the HgT21 strain.** Based on the metal-resistance operons identified in HgT21, resistance to several metal ions ($Hg^+/Zn^{2+}/Cu^{2+}/Co^{2+}/As^{3+}$) was evaluated. The viability of HgT21 bacterial cells in liquid culture exposed to different metal concentrations is shown in Fig. 5. The dose-response curves reveal a gradual decrease in viability when the metal ion concentration increased. Lethal doses of $Hg^+$, $Zn^{2+}$, $Cu^{2+}$, $Co^{2+}$, and $As^{3+}$ were established at 40 $\mu$M, 0.6 mM, 3.0 mM, 420 mM, and 210 mM, respectively.

**Plant growth promotion by the HgT21 strain.** In accordance with the plant growth promotion-associated genes in HgT21, its ability to promote the growth of *Arabidopsis thaliana* seedlings was evaluated. The seedlings were exposed through both direct and distant interactions with HgT21 cells for 8 days. After bacterial exposure, a highly branched root system and leafy shoot were observed compared with the control plants (Fig. 6A). Particularly, an increased number and density of lateral roots were observed in both types of interaction (contact and distant), whereas the inhibition of primary root elongation was evident when roots were in contact with bacteria (Fig. 6A and D). Accordingly, both the wet and dry weights of the plants exposed to direct and distant interactions with HgT21 cells were also increased compared to the control (Fig. 6B and C).

**Indoleacetic acid (IAA) production by HgT21.** Due to the observed ability of the HgT21 strain to induce qualitative and quantitative changes in the growth of *A. thaliana* plants, the production of IAA (a phytohormone commonly produced by PGPB) was evaluated. IAA synthesis by the HgT21 strain was demonstrated through the Salkowski assay, which revealed that 3 mg/L IAA was produced from tryptophan (Fig. 7).

**Susceptibility of the HgT21 strain to antibiotics.** Based on the results of the genome analysis, the HgT21 strain possesses antibiotic resistance-associated genes. To determine the antibiotic multiresistance of HgT21, which could represent a risk for antibiotic resistance dissemination in natural environments, the antibiotic susceptibility of HgT21 was evaluated. The results in Table 2 reveal resistance to some beta-lactam antibiotics (ampicillin, dicloxacillin, and penicillin and the cephalosporins cefotaxime, ceftazidime, and cefuroxime) and optochin (a hydroquinine derivative). Susceptibility to beta-lactam cephalosporins (cephalothin, cefepime,

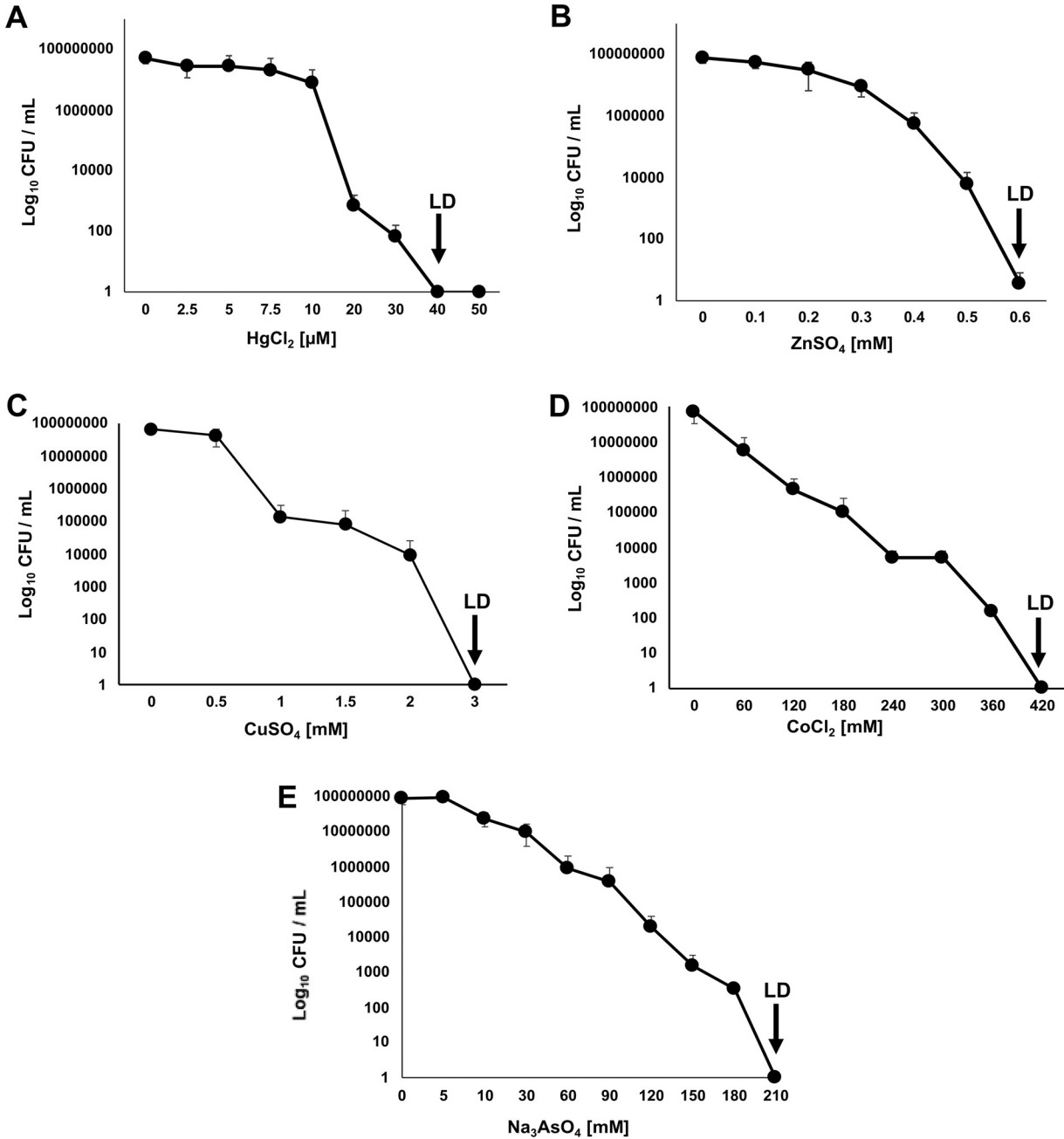

**FIG 5** Lethal doses of different metal ions in HgT21 cells. (A to E) Liquid cultures of exponentially growing cells were exposed to increasing concentrations of mercury (A), zinc (B), copper (C), cobalt (D), and arsenic (E) salts to establish dose-response curves and determine the lethal dose (LD) of each metal tested (black arrows).

and ceftriaxone) and to different groups of antibiotics, including vancomycin (glycopeptide), nitrofurantoin (sulfamide), erythromycin (macrolide), levofloxacin, pefloxacin and ciprofloxacin (quinolones), gentamicin, amikacin and netilmicin (aminoglycosides), polymyxin and bacitracin (polypeptides), novobiocin (aminocumarine), tetracycline, chloramphenicol, and trimethoprim/sulfamethoxazole, was also observed.

## DISCUSSION

In recent decades, sustainable agriculture for safe food production has become an important challenge for soil chemistry, plant biology, agricultural and environmental engineering, and soil microbiology due to the increased demand for healthy safe

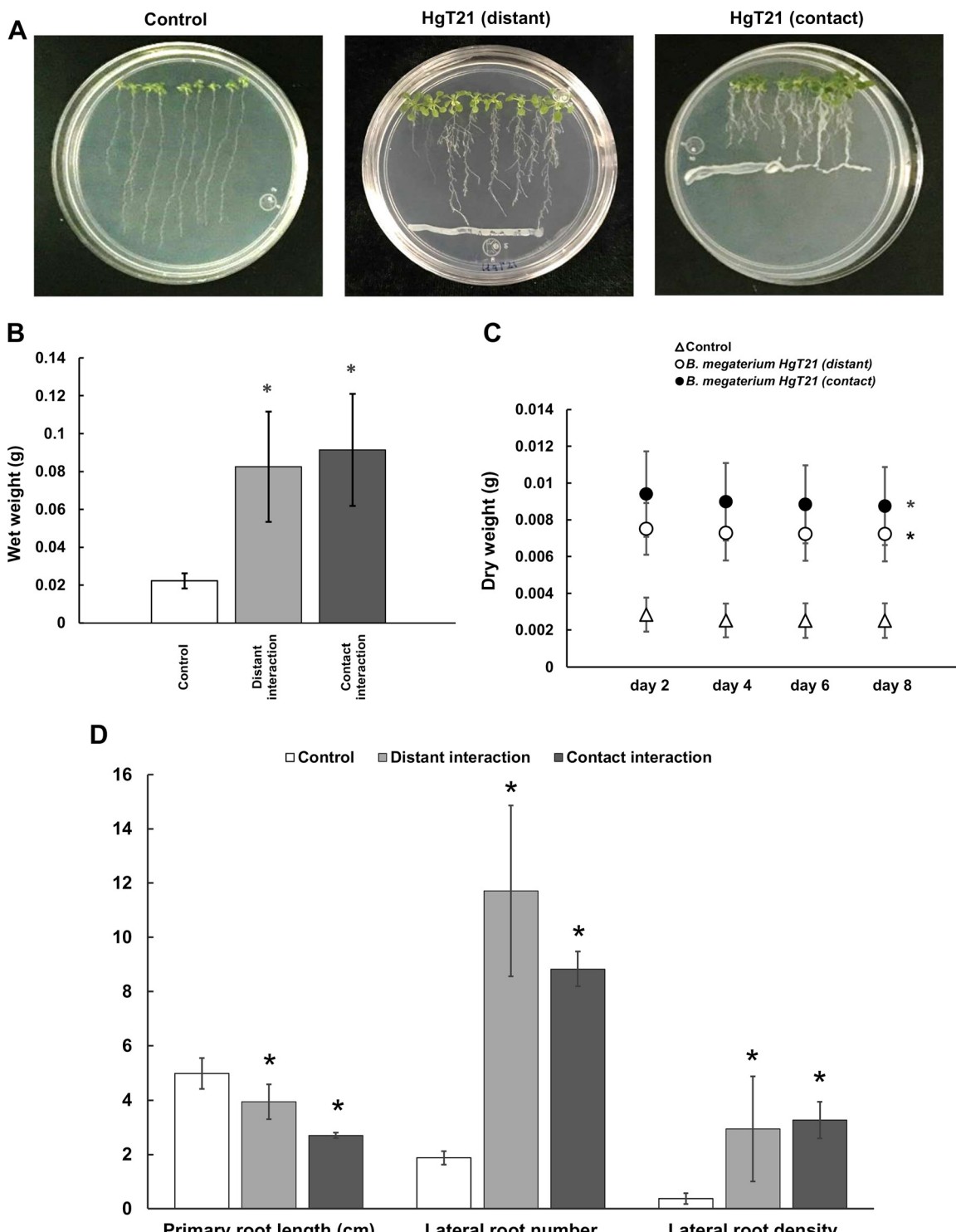

**FIG 6** Growth promotion of *Arabidopsis thaliana* by the HgT21 strain. (A) Germination and growth of *A. thaliana* in the absence (control), distant root interaction, and contact root interaction with *B. megaterium* HgT21. (B) Wet weight of *A. thaliana* plants after 8 days of growth in the presence of HgT21 cells. (C) Dry weight of *A. thaliana* plants exposed to HgT21 cells after 2, 4, 6, and 8 days of dehydration. (D) Primary root length and number and density of lateral roots of *A. thaliana* plants after 8 days of growth in the presence of HgT21 cells. Asterisks (*) indicate significant differences determined by ANOVA ($P < 0.05$).

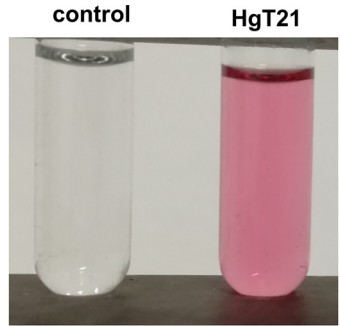

**control**   **HgT21**

**Indolacetic acid (IAA) production**

| Strain | 1 | 2 | 3 | Mean ± SD |
|---|---|---|---|---|
| *B. megaterium* HgT21 | 3,015 µg/mL | 3,058 µg/mL | 3,407 µg/mL | 3,16 ± 0,214 |

**FIG 7** Indole acetic acid (IAA) production by the HgT21 strain. IAA production was observed as a pink-colored supernatant of the HgT21 culture in the presence of L-tryptophan and Salkowski reagent. Three independent experiments of quantitative IAA production were carried out spectroscopically ($OD_{530}$). The mean and standard deviation of determinations are shown.

products (11). Heavy metal pollution in agricultural soils leads to the accumulation of toxic metals in plants and, consequently, their integration into the food chain. Thus, the toxicity of heavy metals for most living organisms at relatively lower concentrations (at the level of ppb or ppm) represents a threat to human health, safe food production, and maintenance of microbial diversity in soils (1, 34). The development of strategies for the recovery of degraded and metal-contaminated agricultural soils could be based on bioremediation approaches using beneficial microorganisms such as metal-resistant PGPB and natural pathogen antagonists as an alternative for metal immobilization or as a replacement for chemical fertilizers and pesticides to guarantee sustainable agriculture (12, 14, 15). In this context, the search for soil plant-associated PGPB and metal biotransformation bacteria is important for the establishment of sustainable strategies for agricultural soil restoration.

In this work, a metal-resistant PGPB was isolated and characterized. Based on 16S and complete-genome sequence analyses, the isolated HgT21 was identified as *Bacillus megaterium*. The phylogenetic tree based on the genome analysis shows that *B. megaterium* HgT21 is closely related to *B. megaterium* DSM319 and is grouped within the *B. aryabhattai-B. megaterium* clade. In addition, the average nucleotide identity (ANI) values (>95 to 96%) between the *B. aryabhattai* and *B. megaterium* strains (Fig. S1) support the proposition of Narsing Rao et al. to reclassify *B. aryabhattai* as the later heterotypic synonym of *B. megaterium* de Bary 1884 (35) due to the ANI values being higher than the recognized threshold values for bacterial species delineation (36). As in *B. megaterium* DSM319, no plasmid sequences were identified; however, a low GC content was observed in some contigs (6 and 8 to 11), and the presence of active prophages (contigs 10 and 11; Fig. 2) and higher coverage (>80×; contigs 6, 8, 10, and 11) suggest the integration of plasmid-borne genes into genomic DNA and/or the exchange of genes between plasmids and chromosomes. This notion is supported by studies that provide evidence of extensive gene transfer between the plasmids of *B. megaterium* QMB1551 and its own chromosome and those of the plasmid-less *B. megaterium* DSM319 (37). According to genotype identification, the morphology of the HgT21 cells correlates with the *B. megaterium* phenotype initially described by De Bary in 1884 (Gram-positive, approximately 8 to 10 $\mu$m size, and spore-forming), and the identification was also corroborated based on biochemical characteristics reported for *B. megaterium* (27).

*Bacillus megaterium* is widely recognized as a soil bacterium with potential industrial applications due to its ability to utilize different carbon sources, grow in a wide

**TABLE 2** Key genes involved in metabolism, auxin biosynthesis, phosphate solubilization, and antibiotic resistance in HgT21

| Process/activity | No. of Contig | ID | Enzyme |
|---|---|---|---|
| **Carbohydrate and polyalcohol metabolism** | | | |
| D-Raffinose | 3 | 4652 | Alpha-galactosidase (EC 3.2.1.22) |
| | 15 | 330 | Alpha-galactosidase (EC 3.2.1.22) |
| D-xylose | 3 | 3729 | Xylose isomerase (EC 5.3.1.5) |
| | 3 | 3730 | Xylulose kinase (EC 2.7.1.17) |
| D-galactose | 3 | 4652 | Alpha-galactosidase (EC 3.2.1.22) |
| | 15 | 330 | Alpha-galactosidase (EC 3.2.1.22) |
| | 3 | 3772 | Beta-galactosidase (EC 3.2.1.23) |
| | 3 | 4022 | Beta-galactosidase (EC 3.2.1.23) |
| D-ribose | 2 | 2867 | Ribose 5-phosphate isomerase A (EC 5.3.1.6) |
| | 2 | 3551 | Ribose 5-phosphate isomerase B (EC 5.3.1.6) |
| | 4 | 4918 | Ribose 5-phosphate isomerase B (EC 5.3.1.6) |
| Lactose | 3 | 3772 | Beta-galactosidase (EC 3.2.1.23) |
| | 3 | 4022 | Beta-galactosidase (EC 3.2.1.23) |
| | 10 | 16 | D-Tagatose 3-epimerase |
| Saccharose/sucrose | 5 | 5078 | 6-Phosphogluconolactonase (EC 3.1.1.31) |
| | 1 | 1987 | 6-Phosphogluconate dehydrogenase, decarboxylating (EC 1.1.1.44) |
| | 2 | 2601 | 6-Phosphogluconate dehydrogenase, decarboxylating (EC 1.1.1.44) |
| | 2 | 3187 | 6-Phosphogluconate dehydrogenase, decarboxylating (EC 1.1.1.44) |
| | 2 | 2867 | Ribose 5-phosphate isomerase A (EC 5.3.1.6) |
| | 2 | 3551 | Ribose 5-phosphate isomerase B (EC 5.3.1.6) |
| | 4 | 4918 | Ribose 5-phosphate isomerase B (EC 5.3.1.6) |
| | 991 | 1443 | Sucrose-6-phosphate hydrolase (EC 3.2.1.B3) |
| | 1 | 1534 | Sucrose-6-phosphate hydrolase (EC 3.2.1.B3) |
| | 20 | 2473 | Sucrose-6-phosphate hydrolase (EC 3.2.1.B3) |
| | 21 | 2475 | Sucrose-6-phosphate hydrolase (EC 3.2.1.B3) |
| | 2 | 3492 | Sucrose-6-phosphate hydrolase (EC 3.2.1.B3) |
| | 2 | 3500 | Sucrose-6-phosphate hydrolase (EC 3.2.1.B3) |
| | 2 | 3153 | Phosphoenolpyruvate-protein phosphotransferase of PTS system (EC 2.7.3.9) |
| D-Maltose | 17 | 377 | Maltose *O*-acetyltransferase (EC 2.3.1.79) |
| | 2 | 2505 | Maltose operon transcriptional repressor MalR, LacI family |
| | 3 | 4173 | Maltose *O*-acetyltransferase (EC 2.3.1.79) |
| | 3 | 4272 | Maltose/maltodextrin ABC transporter, substrate binding periplasmic protein MalE |
| | 9 | 5741 | Maltose *O*-acetyltransferase (EC 2.3.1.79) |
| D-Trehalose | 2 | 2802 | PTS system, trehalose-specific IIB component (EC 2.7.1.69)/PTS system, trehalose-specific IIC component (EC 2.7.1.69) |
| | 2 | 2803 | Trehalose-6-phosphate hydrolase (EC 3.2.1.93) |
| | 2 | 2804 | Trehalose operon transcriptional repressor |
| D-Sorbitol | 3 | 4198 | Sorbitol-6-phosphate 2-dehydrogenase (EC 1.1.1.140) |
| | 6 | 5294 | Glucitol/sorbitol-specific transport protein GutA |
| | 6 | 5295 | Sorbitol dehydrogenase (EC 1.1.1.14) |
| D-Mannitol | 2 | 2862 | Putative transcriptional antiterminator, BglG family/PTS system, mannitol/fructose-specific IIA component (EC 2.7.1.69) |
| | 13 | 241 | Mannitol-1-phosphate 5-dehydrogenase (EC 1.1.1.17) |
| | 13 | 242 | Mannitol operon activator, BglG family |
| | 13 | 243 | PTS system, mannitol-specific IIC component (EC 2.7.1.69)/PTS system, mannitol-specific IIB component (EC 2.7.1.69)/PTS system, mannitol-specific IIA component |
| Starch hydrolysis | 3 | 4365 | Alpha-amylase (EC 3.2.1.1) |
| | 4 | 5002 | Cytoplasmic alpha-amylase (EC 3.2.1.1) |
| **Metabolism of proteins, amino acids and others** | | | |
| Acetoin production | 3 | 3747 | Acetoin dehydrogenase E1 component alpha-subunit (EC 1.2.4.-) |
| | 3 | 3748 | Acetoin dehydrogenase E1 component beta-subunit (EC 1.2.4.-) |
| | 3 | 3749 | Dihydrolipoamide acetyltransferase component (E2) of acetoin dehydrogenase complex (EC 2.3.1) |
| | 3 | 3750 | Dihydrolipoamide dehydrogenase of acetoin dehydrogenase (EC 1.8.1.4) |
| | 3 | 3751 | Transcriptional activator of acetoin dehydrogenase operon AcoR |
| | 3 | 3752 | 2,3-Butanediol dehydrogenase, R-alcohol forming, (R)- and (S)-acetoin-specific (EC 1.1.1.4) |
| Casein hydrolysis | 1 | 1494 | ATP-dependent Clp protease proteolytic subunit (EC 3.4.21.92) |
| | 1 | 2167 | ATP-dependent Clp protease proteolytic subunit (EC 3.4.21.92) |
| | 4 | 4812 | ATP-dependent Clp protease proteolytic subunit (EC 3.4.21.92) |

**TABLE 2** (Continued)

| Process/activity | No. of Contig | ID | Enzyme |
|---|---|---|---|
| Cytochrome oxidase | 1 | 1256 | Cytochrome oxidase biogenesis protein Sco1/SenC/PrrC, putative copper metallochaperone |
| Lysine descarboxylase | 14 | 256 | Arginine decarboxylase (EC 4.1.1.19)/Lysine decarboxylase (EC 4.1.1.18) |
| | 3 | 3933 | Lysine decarboxylase family |
| Ornithine descarboxilase | 2 | 2528 | Ornithine carbamoyltransferase (EC 2.1.3.3) |
| | 2 | 2507 | Ornithine aminotransferase (EC 2.6.1.13) |
| Arginine dihydrolase 1 | 2 | 3203 | Arginine decarboxylase (EC 4.1.1.19) |
| | 14 | 256 | Arginine decarboxylase (EC 4.1.1.19)/lysine decarboxylase (EC 4.1.1.18) |
| Alanine arylamidase | 1 | 686 | *N*-acetylmuramoyl-L-alanine amidase (EC 3.5.1.28) |
| | 1 | 1212 | *N*-acetylmuramoyl-L-alanine amidase (EC 3.5.1.28) |
| | 1 | 1688 | *N*-acetylmuramoyl-L-alanine amidase |
| | 1 | 2095 | *N*-acetylmuramoyl-L-alanine amidase (EC 3.5.1.28) |
| | 1 | 2201 | *N*-acetylmuramoyl-L-alanine amidase (EC 3.5.1.28) |
| | 2 | 2903 | *N*-acetylmuramoyl-L-alanine amidase |
| | 3 | 3925 | *N*-acetylmuramoyl-L-alanine amidase (EC 3.5.1.28) |
| | 3 | 3928 | *N*-acetylmuramoyl-L-alanine amidase (EC 3.5.1.28) |
| | 3 | 4105 | *N*-acetylmuramoyl-L-alanine amidase |
| Tyrosine arylamidase | 1 | 2373 | Aminopeptidase S (Leu, Val, Phe, Tyr preference) (EC 3.4.11.24) |
| Beta galactosidase | 3 | 4022 | Beta-galactosidase (EC 3.2.1.23) |
| | 3 | 3772 | Beta-galactosidase (EC 3.2.1.23) |
| Alpha galactosidase | 15 | 330 | Alpha-galactosidase (EC 3.2.1.22) |
| | 3 | 4652 | Alpha-galactosidase (EC 3.2.1.22) |
| Alpha glucosidase | 2 | 3289 | Alpha-glucosidase (EC 3.2.1.20) |
| | 1 | 1533 | Alpha-glucosidase (EC 3.2.1.20) |
| *N*-acetyl-D-glucosaminidase | 18 | 386 | Spore cortex-lytic enzyme, *N*-acetylglucosaminidase SleL (EC 3.2.1.-) |
| | 2 | 3351 | Spore peptidoglycan hydrolase (*N*-acetylglucosaminidase) (EC 3.2.1) |
| Urease | 1 | 2274 | Urease gamma subunit (EC 3.5.1.5) |
| | 1 | 2275 | Urease beta subunit (EC 3.5.1.5) |
| | 1 | 2276 | Urease alpha subunit (EC 3.5.1.5) |
| | 1 | 2277 | Urease accessory protein UreE |
| | 1 | 2278 | Urease accessory protein UreF |
| | 1 | 2279 | Urease accessory protein UreG |
| | 1 | 2280 | Urease accessory protein UreD |
| Auxin biosynthesis | | | |
| IAA synthesis | 2 | 3619 | Acetolactate synthase large subunit (EC 2.2.1.6) |
| | 2 | 2637 | Alpha-acetolactate decarboxylase (EC 4.1.1.5) |
| | 2 | 2638 | Acetolactate synthase, catabolic (EC 2.2.1.6) |
| | 1 | 614 | Acetolactate synthase large subunit (EC 2.2.1.6) |
| | 1 | 615 | Acetolactate synthase small subunit (EC 2.2.1.6) |
| | 3 | 3752 | 2,3-butanediol dehydrogenase, R-alcohol forming, (R)- and (S)-acetoin-specific (EC 1.1.1.4) |
| | 1 | 1836 | 2,3-butanediol dehydrogenase, R-alcohol forming, (R)- and (S)-acetoin-specific (EC 1.1.1.4) |
| | 3 | 3691 | Glycerol dehydrogenase (EC 1.1.1.6) |
| Phosphate solubilization | | | |
| | 2 | 3621 | Glucose dehydrogenase (pyrroloquinoline-quinone) |
| | 2 | 3622 | Glucose dehydrogenase (pyrroloquinoline-quinone) |
| | 7 | 5440 | L-lactate dehydrogenase (EC 1.1.1.27) |
| | 2 | 3087 | Predicted L-lactate dehydrogenase, Fe-S oxidoreductase subunit YkgE |
| | 2 | 3088 | Predicted L-lactate dehydrogenase, iron-sulfur cluster-binding subunit YkgF |
| | 2 | 3089 | Predicted L-lactate dehydrogenase, hypothetical protein subunit YkgG |
| | 1 | 2185 | Citrate synthase (si) (EC 2.3.3.1) |
| | 1 | 544 | Citrate synthase (si) (EC 2.3.3.1) |
| | 3 | 4390 | Manganese-dependent inorganic pyrophosphatase (EC 3.6.1.1) |
| | 1 | 548 | Alkaline phosphatase synthesis transcriptional regulatory protein PhoP |
| | 1 | 2268 | Alkaline phosphatase |
| | 2 | 2484 | Alkaline phosphatase-like protein |
| | 2 | 3067 | Alkaline phosphatase (EC 3.1.3.1) |
| | 2 | 3367 | Alkaline phosphatase (EC 3.1.3.1) |
| | 2 | 3581 | Alkaline phosphatase like protein |
| | 4 | 5006 | Alkaline phosphatase (EC 3.1.3.1) |

**TABLE 2** (Continued)

| Process/activity | No. of Contig | ID | Enzyme |
|---|---|---|---|
| Antibiotic resistance | | | |
| | 1 | 478 | Metallo-beta-lactamase family protein |
| | 1 | 1333 | Metallo-beta-lactamase family protein |
| | 1 | 1899 | Beta-lactamase class A |
| | 2 | 2615 | Metallo-beta-lactamase family protein |
| | 2 | 3330 | Beta-lactamase (EC 3.5.2.6) |
| | 2 | 3410 | Beta-lactamase (EC 3.5.2.6) |
| | 2 | 3473 | Metallo-beta-lactamase family protein |
| | 2 | 3657 | Beta-lactamase class C and other penicillin binding proteins |
| | 3 | 4280 | Metal-dependent hydrolases of the beta-lactamase superfamily I; PhnP protein |
| | 4 | 5011 | Zn-dependent hydrolase (beta-lactamase superfamily) |
| | 7 | 5353 | Metal-dependent hydrolases of the beta-lactamase superfamily I; PhnP protein |

range of temperatures (3 to 45°C), produce proteases, promote plant growth, and antagonize plant pathogens for biocontrol (38–40). In agreement with these reports, genome annotation based on protein prediction revealed that *B. megaterium* HgT21 contains approximately 264 genes related to the production of bacteriocin and antibacterial peptides, resistance to antibiotic and toxic compounds, auxin biosynthesis, and stress response. Moreover, the biochemical profile of *B. megaterium* HgT21 reveals its ability to use diverse carbon sources (Table 1). These results are consistent with key enzyme genes encountered in its genome, which are involved in the metabolism of these carbohydrates and polyalcohols (Table 2). Additional enzymatic activities detected in *B. megaterium* HgT21 are also correlated with the presence of key genes involved in these enzymatic activities (Table 2), except for urease; in contrast to the biochemical data, urease was identified in the *B. megaterium* HgT21 genome, suggesting dysfunction of this enzyme.

According to the recognition of the *Bacillus* genus as PGPB, genes involved in one of the three metabolic pathways described for the synthesis of indole-3-acetic acid (IAA), a nonvolatile phytohormone involved in plant growth promotion (41), were found in *B. megaterium* HgT21. The production of IAA by PGPB is related to the synthesis pathway of tryptophan. In *B. megaterium* HgT21, the genes required for IAA synthesis through the chorismate pathway were identified (Table 2) and involve the conversion of chorismate to tryptophan (by enzymes encoded in the *trpABCDEG* operon) and its subsequent conversion to IAA by nitrilase activity. These results suggest that enzymes encoded by genes of the chorismate pathway could be involved in IAA production by *B. megaterium* HgT21. Moreover, genes involved in the synthesis of volatile organic compounds (VOCs) such as acetoin and butanediol were also identified; these genes encode enzymes that convert pyruvate to acetoin (acetolactate synthase and acetolactate decarboxylase) and acetoin to 1,3-butanediol (2,3-butanediol dehydrogenase [BDH] and glycerol dehydrogenase [GDH]) (42). Although the activity of these enzymes was not demonstrated in this work, the presence of their encoding genes in the HgT21 genome could be related to the increased growth of *A. thaliana* seedlings observed after its exposure to both distant and contact interactions with *B. megaterium* HgT21. The phenotype observed in *A. thaliana* after the interaction with *B. megaterium* HgT21 correlates with previous reports that describe the increase of lateral roots, decreasing cell elongation in primary root, and root hair density increase upon colonization (43). In relation to the phenotype observed, it has been described that some PGPB indirectly reduce the effect of ethylene production (an inhibitor of primary growth root) through the production of the ACC (1-aminocyclopropane-1-carboxylate) deaminase, which degrades ethylene, and in consequence induces primary root elongation (44). However, the gene encoding the ACC deaminase was not found in *B. megaterium* HgT21; this could explain the inhibition of the root growth when colonization is established.

According to reports that demonstrate that the production of VOCs by rhizobacteria induces the elongation of lateral roots and root hairs of *A. thaliana* seedlings (45), the

production of acetoin and 2,3-butanediol in HgT21 could be responsible for the observed elongation of lateral roots and root hairs of seedlings; however, this must be experimentally demonstrated, and future experiments must be conducted to evaluate the contribution of VOCs and diffusible compounds to growth promotion. Additionally, the group of genes involved in phosphate solubilization (41, 46) that were identified in *B. megaterium* HgT21 reveal its potential as a phosphate solubilizer bacterium.

*Bacillus megaterium* is also recognized for its adaptation and resistance to stressful environmental conditions such as saline or acidic soils and soil contamination with heavy metals and other xenobiotics (40, 47, 48). Accordingly, *B. megaterium* HgT21 showed high tolerance to $Hg^{II+}$ ions, and the MIC values were higher in solid (975 $\mu$M $HgCl_2$) than in liquid (75 $\mu$M $HgCl_2$) medium, presumably due to the major interaction of the bacterial cell surface with metal ions in liquid medium (49, 50). This result correlates with observations in mercury-resistant *Bacillus* species isolated from Minamata Bay, Japan, a highly mercury-contaminated site, which have MIC values from 80 to 320 $\mu$M $HgCl_2$, particularly the well-characterized mercury-resistant strain *B. megaterium* MB1 (MIC of 80 $\mu$M $HgCl_2$). These values are significantly higher than those of the mercury-sensitive strain (no *mer* determinants) *B. megaterium* WH20 (MIC of 10 $\mu$M $HgCl_2$) (16, 22). Genomic analysis revealed that *B. megaterium* HgT21 contains a variety of heavy metal resistance-associated genes and/or complete putative operons, including the *merRETPA* operon and the *merB3* gene, which confer a wide spectrum of mercury resistance (51). The mercury resistance system in HgT21 showed a gene arrangement identical to that of the *mer*-like determinant in Tn*5083* of *B. megaterium* MK64-1 (Kamchatka, Russia), which has been proposed to be a derivative of *mer* determinants reported in Tn*5084* of *Bacillus cereus* RC607 (Boston, USA) and Tn*MERI1* of *B. megaterium* MB1 (Minamata, Japan) (21, 22, 52–54). These results support the idea that recombination events of transposition gene exchange could be an important contribution to the evolution of $Hg^{II+}$-resistant transposons in Gram-positive bacteria and the worldwide horizontal dissemination of the class II Tn*MERI1*-like transposons across bacterial species and geographical barriers (22, 55).

*Bacillus megaterium* HgT21 also contains a variety of metal(oid) resistance systems, which include well-described genes and operons for arsenic, copper, cadmium, zinc, cobalt, and tellurium resistance. Two arsenic resistance-like operons were identified in HgT21. First, the canonical operon *arsRBC* (contig 8), initially described in *Staphylococcus aureus* (pI258 plasmid), correlates with its wide distribution in the plasmids and chromosomes of *Bacteria* and *Archaea* from different origins (32). However, in *Bacillus* species, only variants of this operon have been reported, as well as the *arsR2-orf2-arsB-arsC2* operon in the chromosome of *Bacillus* sp. strain CDB3 and the *arsR-orf2-acr3-arsC* operon in the "Tn (skin element)" of *Bacillus subtilis* JH642 (31, 56). The second arsenic resistance-like operon (*arsRBCDA*) identified (contig 2) showed an arrangement similar to that of the commonly reported operon *arsRDABC* (57) and was identical to those reported in the chromosome of *Bacillus* sp. CDB3 (31). The presence of two arsenic resistance operons, extra copies of *arsR* (upstream from *arsRBC* and *arsRBCDA*) and *acr3* (upstream from *arsRBCDA*), correlates with the wide complexity and variety of gene configurations in *ars* clusters reported in prokaryotes (32, 58). The enzymes encoded by the arsenate resistance-like operons in HgT21 could be responsible for the observed arsenate resistance in this strain, which is similar to the most arsenic-resistant *Bacillus* species (56, 59, 60). The presence of the *acr3* gene (*arsB* homolog) in *Bacillus megaterium* HgT21 contrasts with studies about the prevalence of *acr3* in different orders of *Bacteria*, which revealed that *acr3* is mainly present in *Burkholderiales*, *Actinobacteria*, and *Alphaproteobacteria*, whereas *arsB* is prevalent in *Firmicutes* and *Gammaproteobacteria* (61). Taken together, the results support studies that revealed that the presence of multiple and redundant *ars* genes in bacteria exposed to selective pressure is the result of gene duplication via horizontal transfer (32).

The arrangement of the three copper resistance-associated genes in *B. megaterium* HgT21 (*csoR-copZ-copA*) is identical to the copper resistance system described in *B. subtilis* (33), which includes the transcriptional regulator CsoR (a widespread copper-inducible repressor distributed in Gram-positive bacteria and *Proteobacteria* that regulates expression of the *copZA* operon), the copper chaperone CopZ, and the copper-efflux ATPase CopA (62). The presence of

this copper-like resistance system in *B. megaterium* HgT21 correlates with the identification of this system in approximately half of the members of *Firmicutes*, including the *Bacillales* "copper users," for which copper is essential for aerobic respiration, acting as a cofactor in terminal enzymes of the aerobic pathway (63). In copper users, the *csoR-copZA* system confers resistance to high levels of copper (33), as was observed in *B. megaterium* HgT21 (3 mM $CuSO_4$), compared with the copper resistance reported in both Gram-positive and Gram-negative bacteria (3.5 to 5.5 mM $CuSO_4$) (12, 33, 64). The presence of two identical copper resistance systems at different locations in the genome suggests duplication of this system associated with mobile elements.

Three genes associated with resistance to different metal cations ($Zn^{2+}$, $Cd^{2+}$, $Pb^{2+}$, and $Co^{2+}$) were identified in *B. megaterium* HgT21. *cadA* encodes a cadmium-translocating P-type ATPase (CadA), a homologous protein of ZntA, which is considered a multipurpose metal-exporting pump for the extrusion of $Zn^{2+}$, $Cd^{2+}$, $Ag^{2+}$, and $Pb^{2+}$ (65). The *cadA* gene is a component of the well-described cadmium resistance operon *cadAC* encountered in Gram-positive bacteria, including *S. aureus* (plasmid pI258), *Listeria monocytogenes*, *B. megaterium*, and other *Bacillus* species (24, 66–68). In the CadAC system, the expression of *cadA* is tightly controlled by the regulatory protein CadC, a $Cd^{2+}/Pb^{2+}/Zn^{2+}$ responsive repressor encoded by *cadC* located downstream of *cadA* (67). Interestingly, although *cadA* was located in *B. megaterium* HgT21, *cadC* was not found; instead, a hypothetical protein was located upstream of *cadA*. However, in the opposite direction, near and upstream of *cad*A, the genes *ars*R (encoding an ArsR family transcriptional regulator) and *czcD* (encoding a membrane-bound protein member of the metal-diffusion facilitator, cation diffusion facilitator (CDF) subfamily) were found. This suggests that in the absence of CadC (a homodimeric repressor that belongs to the ArsR/SmtB family of metalloregulatory proteins), ArsR could act as a transcriptional regulator of *cadA* (67). With respect to *czc*D, it has been reported that CzcD is a heavy metal transporter involved in the regulation of the *czc* (cadmium, zinc, cobalt) resistance system described in *Ralstonia* sp. strain CH34 (*Cupriavidus metallidurans* CH34) (69, 70); however, the absence of additional *czc* genes in *B. megaterium* HgT21 suggests that, rather than acting as a regulator protein, CzcD could mediate resistance against $Zn^{2+}/Co^{2+}/Ni^{2+}/Cd^{2+}$ through an antiporter mechanism catalyzing the active efflux of divalent ions in exchange for $K^+/H^+$, as described in *Bacillus subtilis* (71, 72). Thus, *czcD* could play a physiological role related to the maintenance of metal homeostasis, as has been proposed for members of CDF family, such as CzcD in *Streptococcus pneumoniae*, ZRC-1 in *S. cerevisiae* and Znt-1 and Znt-2 in mammals (73–75). In agreement with this result, the presence of *czcD* without other components of the *czc* and *cadAC* operon was described in *Bacillus megaterium* (26) and *Bacillus subtilis* (71). In *Bacillus paranthracis*, both *czcD* and *cadA* are present; however, the gene arrangement and genome location have not been described (23). Thus, to our knowledge, this is the first report of a genic arrangement that includes *cadA*, *arsR*, and *czcD*, each corresponding to three different metal-resistance systems (*czc*, *cadAC*, *ars*). CadA and CzcD are involved in the extrusion of divalent metal ions, suggesting that this chimeric system probably arises for the resistance to and/or homeostasis maintenance of divalent metal cations ($Zn^{2+}$, $Cd^{2+}$, $Pb^{2+}$, and $Co^{2+}$) in *B. megaterium* HgT21, which showed $Zn^{2+}$ resistance (0.6 mM) comparable to that reported for zinc-resistant bacteria (0.1 to 1.0 mM) (65, 71); however, this must be experimentally demonstrated. Interestingly, the $Co^{2+}$ resistance was extremely high (420 mM) relative to the $Co^{2+}$ resistance reported in bacteria (0.425 to 8 mM $CoCl_2$) (70, 76), and the mechanism involved in this unprecedented level of $Co^{2+}$ resistance is under investigation.

A set of genes associated with tellurium resistance (Te^R) was also identified in *B. megaterium* HgT21. Te^R determinants are a group of ubiquitous genes distributed across phylogenetically diverse taxa in bacteria isolated from diverse environments (77). Based on the high sequence similarity, it has been proposed that Te^R genes arose from a common ancestor that evolved as the result of adaptation to ancient metal-rich environments or through horizontal transfer events (77, 78). A variety of Te^R genes have been identified in plasmids and chromosomal genomic islands (GIs) of pathogenic and extreme-environment isolated bacteria; however, the function of proteins encoded by these genes remains unclear

(77, 79). The operon *terZABCDE* is commonly reported in bacteria; however, the minimal fragment required to confer tellurite resistance is *terBCDE*, whereas the complete operon also provides phage resistance and pore-forming colicins, explaining its maintenance in a variety of pathogenic bacteria for which tellurite resistance is probably not necessary (80). Moreover, comparative genomics revealed that *ter* genes are functionally linked to enzymes involved in DNA processing and repair and induced by $H_2O_2$ and superoxide, suggesting their association with the general stress response caused by ROS production or tellurite ions (81, 82). In this context, *telA*, *yceG*, and the three tandem copies of *terD* in *B. megaterium* HgT21 could be associated with different functions: (i) tellurite resistance, considering that it was isolated from a metal-polluted environment and considering the role of TelA in tellurite resistance in *Rhodobacter sphaeroides* (83), and/or (ii) to deal with hostile environmental conditions related to ROS generation, considering that *yceG* and *terD* have been associated with the cellular defense against oxidative stress in pathogenic bacteria (82, 84). However, the function of enzymes encoded in tellurite resistance-related genes in *B. megaterium* HgT21 must be experimentally demonstrated.

Due to bacterial metal resistance being associated with antibiotic resistance (85, 86), the antibiotic susceptibility in *B. megaterium* HgT21 was assessed. Susceptibility to most of the antibiotic families was observed, suggesting a low risk of antibiotic resistance transfer in the environment. Resistance to some beta-lactams correlates with the presence of genes associated with beta-lactam resistance. Optochin resistance could probably be produced by a lack of recognition of the target site of this antibiotic (membrane ATPase F0F1), which can be altered by mutations in the ATPase F0F1 gene (87). The resistance to beta-lactams, bacteriocins, and antibacterial peptides in *B. megaterium* HgT21 could be advantageous in its natural hostile environment, in which competence with other microorganisms could be crucial to survive.

The *Bacillus* species in the soil habitat are well recognized as "zymogenous" bacteria and ecologically have been defined as "r-strategists," which means that they can grow quickly when the nutrient supply is abundant, as in the rhizosphere, and possess a high colonization and competitive ability. Importantly, quorum sensing-mediated processes in *Bacillus* species, such as endospore and biofilm formation, constitute an important survival strategy in soil under nutrient-limited conditions and hostile environments; its high adaptability to the environment is evidenced by its ubiquity in both nonextreme and extreme soils (88). In this context, the genetic information of this bacterium and the preliminary physiological trait analysis (metal multiresistance, plant growth promotion, IAA production, use of carbon sources) suggest a high adaptability of *B. megaterium* HgT21 to adverse environments such as metal-contaminated soils and its potential as a PGPB through phytohormone and VOC production, phosphate solubilization, and antibiotic production.

Considering the essential characteristics that help to define new PGPB as biofertilizers (high rhizosphere competence, ability to increase plant biomass, long-term survival, plant-beneficial physiological traits, a lack of risk factors for human and environmental health, and high tolerance to environmental stresses encountered in soil/plant systems) (88), the genetic information of the HgT21 strain suggests that it could be suitable for use as a biofertilizer in the promotion of plant growth in metal-contaminated environments due to its ability to produce/perform quorum sensing-related processes, endospores, flagella, siderophores, metabolism of a variety of organic compounds, IAA production, and resistance to a variety of metals. However, *in vivo* experiments under more realistic conditions must be conducted to demonstrate that it can be used as a good biofertilizer. Moreover, due to its sporulating, nonpathogenic, and free-endotoxin nature, *B. megaterium* HgT21 could be a good candidate for biotechnology applications in the food and pharmaceutical industries, among others (38).

## MATERIALS AND METHODS

**Isolation of the mercury-tolerant HgT21 bacterial strain.** The mercury-tolerant bacterial strain HgT21 was isolated from the rhizosphere of Fabaceae plants (*Dalea bicolor*) inhabiting mining tails located in

Zacatecas, Mexico (22°47′01.4″ N, 102°36′21″ W and 2,426 meters above sea level [m.a.s.l.]). To test the HgT21 tolerance to mercury ions ($Hg^{II+}$) in solid medium, the MIC was determined. Briefly, LB agar was supplemented with $HgCl_2$ at final concentrations of 0, 1, 10, 100, 200, 400, 800, 850, 900, 925, 950, 975, and 1,000 $\mu$M and poured into petri dishes. After agar solidification, the plates were inoculated with $1 \times 10^5$ colony forming units (CFU) and spread onto the agar surface and incubated at 37°C until visible growth of bacteria was detected. The concentration at which no growth was detected was reported as the MIC in solid medium. Mercury tolerance in liquid medium was also determined, by assessing the MIC value as follows: overnight cultures were diluted 1:100 in fresh LB medium supplemented with $HgCl_2$ at final concentrations of 0, 1, 10, 25, 50, 75, and 100 $\mu$M and grown at 37°C and 200 rpm for 8 h (when the end of exponential growth was reached in cultures not exposed to mercury ions), and then the optical density was spectroscopically determined at 600 nm ($OD_{600}$). Three independent experiments were carried out.

**Morphological and biochemical characterization of the HgT21 strain.** This analysis was based on colony characteristics, Gram staining, motility, and biochemical properties such as $H_2S$ production, acid and acetylmethylcarbinol production from glucose, use of citrate and mannitol as a carbon source, and the presence of enzymatic activities such as urease, lysine and ornithine decarboxylases, tryptophanase, cytochrome oxidase, catalase, and amylase (89). Based on this characterization, the HgT21 strain was identified according to Bergey's Manual of Determinative Bacteriology (27). An additional test for the biochemical analysis was performed using an automated Vitek 2 system (bioMérieux) with the GP-ID card for Gram-positive bacteria.

**Genome sequencing, *de novo* assembly, and annotation.** Genome sequencing of *Bacillus megaterium* HgT21 was performed at Langebio genomics services (Irapuato, Mexico City) using the $2 \times 300$-bp Illumina MiSeq platform. Raw reads were quality assessed using cutadapt v1.9.1 (90) and Trimmomatic v0.36 (91) to remove adapter sequences and low-quality reads, respectively. *De novo* assembly of the filtered reads was performed using SPAdes v3.9.1 (92) with the –careful parameter for read error correction. The assembled contigs were annotated using the Rapid Annotation using Subsystem Technology (RAST) web server (http://rast.nmpdr.org) (93) using the ClassicRAST annotation scheme, FIGfams v90, automatic error correction, and automatic frameshift correction. Finally, noncoding RNAs (ncRNAs), tRNAs, and rRNAs were searched in the HgT21 assembly using Infernal v1.1.3 (94) against the Rfam database v14.1 (95). Prophage regions were bioinformatically predicted with Prophage Hunter (96). Only regions with scores of >0.8 were classified as active prophage regions. The presence of plasmid sequences was verified using PLACNETw (97) and PlasmidFinder v2.1 (98). Colinear blocks between contigs of the *B. megaterium* HgT21 genome were identified using the SibeliaZ-LCB (99) algorithm with default parameters. Gene and ncRNA densities, colinearity blocks, GC content, and prophage regions were visualized in Circos (100). The gene regions were visualized using the R package gggenes v0.4.1 (https://github.com/wilkox/gggenes).

**Identification of HgT21 and phylogenetic construction.** We used a whole-genome multilocus sequence typing (wgMLST) method (101) to identify the HgT21 strain. Protein sequences of 28 species phylogenetically closely related to HgT21 were used for phylogenetic analysis (Table S1). The selection of representative species was carried out according to their pairwise average nucleotide identity (ANI) values, which were determined by FastANI (102). The species tree was inferred using the STAG algorithm and rooted with the STRIDE algorithm in the OrthoFinder program (103). *Bacillus cereus* B4264 was used as an outgroup.

**Pan- and core-genome analysis.** Gene orthogroups were derived from OrthoFinder analysis. The gene count of all orthogroups was converted to a 0/1 matrix (1 indicates the presence of the gene in the orthogroup, and 0 indicates its absence). The resulting binary matrix was used to generate the gene accumulation curves of the pan- and core genomes using PanGP software (104) with the distance guide method, sample size of 5,000, 100 sample repeats, and 100 amplification coefficients. The species tree and panmatrix were plotted together using the phytools package (105). In addition, the proteins of the pangenome were grouped into one of the three studied cores, *B. megaterium-B. aryabhattai*, outgroup, and all species, after they were mapped to different COGs using the WebMGA server (106) and RPS-BLAST program.

**Metal resistance of the HgT21 strain.** The survival of HgT21 cells exposed to different metal salts was evaluated as a measurement of their resistance to different metal ions. Lethal doses (LDs; concentration at which the bacterial cells were dead) of metal ions were established throughout viability assays. Bacterial cells were exposed to different metal salts, including sodium arsenite, sodium arsenate, mercury chloride, zinc sulfate, copper sulfate, and cobalt chloride. Briefly, 1 mL of HgT21 cells at the mid-log phase of growth in LB medium ($1 \times 10^8$ CFU/mL) was deposited in 24-well polystyrene plates (Nunclon Delta surface, Nunc) and treated with metallic salts at final concentrations of 0, 2.5, 5.0, 7.5, 10, 20, 30, 40, and 50 $\mu$M $HgCl_2$; 0.1, 0.2, 0.3, 0.4, 0.5, and 0.6 mM $ZnSO_4$; 0, 0.5, 1.0, 1.5, 2.0, and 3 mM $CuSO_4$; 0, 60, 120, 180, 240, 300, 360, and 420 mM $CoCl_2$; or 0, 5, 10, 30, 60, 90, 120, 150, 180, and 210 mM $Na_3AsO_4$. The plate was incubated for 2 h at 37°C and 120 rpm. Subsequently, 0.1 mL of each well was taken for serial dilution, and 0.1 mL of each dilution was plated in triplicate onto LB agar and incubated at 37°C for 18 h. The colonies formed were counted and graphically reported as CFU per milliliter (CFU/mL) as a measure of the bacterial viability as a function of the metal ion concentration. Finally, the LD of each metal ion was determined.

**Analysis of plant growth promotion by the HgT21 strain.** Seeds of *Arabidopsis thaliana* ecotype Col-0 were surface-sterilized with 95% (vol/vol) ethanol solution for 5 min and bleach (20% [vol/vol]; equivalent to 0.13 M sodium hypochlorite) for 7 min. Subsequently, the seeds were washed five times with sterile distilled water and stored at 4°C for 48 h in the dark to promote and synchronize germination. The stratified *A. thaliana* seeds were placed and germinated on petri dishes containing 0.2× Murashige and Skoog (MS) medium, pH 5.7, and 1% agar TC (micropropagation grade, PhytoTechnology

Laboratories) in a horizontal arrangement on the upper zone (eight plants in each plate with 2 cm inter-distance) before bacterial inoculation (107). Plates were sealed with plastic film to avoid seed contamination and placed vertically with an angle of 65° in a growth chamber (Percival Scientific) at 22 to 20°C with a photoperiod of 16 h of light and 8 h darkness.

The plant growth promotion of the HgT21 strain on *A. thaliana* was carried out by *in vitro* direct (in contact) and indirect (distant) interactions of plant root tips with bacterial cells as previously described (108). For this analysis, 6-day-old plates with *A. thaliana* seedlings were inoculated by streaking with 10 $\mu$L (approximately $1 \times 10^6$ CFU/mL) of a 24-h bacterial culture of the HgT21 strain using an inoculating loop to draw a straight line over the middle (contact) or the lower zone (distant) of plates containing the *A. thaliana* seedlings. The inoculation of the lower zone avoids the interaction of plant root tips with bacterial cells (distant), whereas the inoculation in the middle zone allows for the interaction between bacterial cells and plant roots (contact). Petri dishes without bacterial inoculation were also prepared and used as controls. After bacterial inoculation, the plates were incubated in a growing chamber at 22 to 20°C with a photoperiod of 16 h of light and 8 h of darkness. Every 2 days, until day 8 postinoculation, photographs of plates were taken to determine primary root length, number and density of lateral roots (density = number of lateral roots/length of primary root) using ImageJ software (http://rsb.info.nih.gov/nih-image/). In addition, seedlings were removed from the plates, weighed, and dried at 80°C in paper bags until they reached a constant weight. The wet and the dry biomass was determined. All experiments were replicated at least three times. The differences in the number and density of lateral roots, primary root length, and the wet and dry weights of plants exposed and not exposed (control) to direct and distant interactions with HgT21 were calculated by performing one-way analysis of variance (ANOVA) in Microsoft Excel with significance set at $P < 0.05$.

**Indoleacetic acid (IAA) production by the HgT21 strain.** A culture of HgT21 at the mid-log phase of growth was prepared in LB medium, and 0.1 mL of this culture (containing $1 \times 10^8$ CFU/mL) was inoculated in 5 mL of M9 minimal medium supplemented with 0.5% glucose and 0.1% L-tryptophan and incubated for 96 h at 37°C and 180 rpm in triplicate. The amount of IAA produced was determined by the colorimetric method described by Gordon and Weber (109).

**Antibiotic resistance of the HgT21 strain.** Resistance to different antibiotics was tested by the disk diffusion method and interpreted according to the manufacturer's instructions for BBL antibiotic SensiDiscs (Becton, Dickinson Microbiology Systems, Cockeysville, MD) or Multidiscos (Bio-Rad, Mexico).

**Data availability.** The raw sequence reads have been deposited in the NCBI Sequence Read Archive (SRA; SRR17752569 and SRR17752570) under BioProject accession number PRJNA800475. The draft genome sequence was submitted to GenBank with the accession number JAKKUX000000000.

## SUPPLEMENTAL MATERIAL

Supplemental material is available online only.
**SUPPLEMENTAL FILE 1**, PDF file, 0.6 MB.

## ACKNOWLEDGMENTS

We gratefully acknowledge the computing time granted by CNS-IPICYT, grant TKII-R2022-LFGO, CONACyT scholarships for L.T.-S. (grant 422855) and I.N.Q.-S. (grant 708204), and COZCyT for publication fee grant.

Conceptualization: J.G-M., L.F.G.-O., L.E.V.-R. Formal analysis: J.G.-M., L.F.G.-O., L.E.V.-R., L.S.-C. Investigation: J.G.-M., L.F.G.-O., L.T.-S., P.R.-N., I.N.Q.-S, L.E.V.-R. Project administration: L.E.V.-R. Resources: R.M.R.-S., L.S.-C., L.E.V.-R. Validation: J.G.-M. and L.F.G.-O. Writing-original draft: J.G.-M., L.F.G.-O., L.E.V.-R. Writing-review and editing: J.G.-M., L.F.G.-O., L.S.-C. and L.E.V.-R.

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
