## [Reviewer comments · Microbiology Spectrum]

Microbiology Spectrum

***Bacillus megaterium* HgT21: A promising metal multiresistant plant growth promoting bacteria for soil bioremediation**

Jesús Guzmán-Moreno, Luis García-Ortega, Lilia Torres-Saucedo, Paulina Rivas-Noriega, Rosa Ramírez-Santoyo, Lenin Sánchez-Calderon, Iliana Quiroz-Serrano, and Luz Vidales-Rodríguez

Corresponding Author(s): Luz Vidales-Rodríguez, UNIVERSIDAD AUTONOMA DE ZACATECAS

Review Timeline:

Submission Date:	February 21, 2022
Editorial Decision:	May 7, 2022
Revision Received:	June 16, 2022
Accepted:	July 26, 2022

Editor: Benjamin Wolfe

Reviewer(s): Disclosure of reviewer identity is with reference to reviewer comments included in decision letter(s). The following individuals involved in review of your submission have agreed to reveal their identity: Nasim Maghbolli Balasjin (Reviewer #1)

Transaction Report:

DOI: <https://doi.org/10.1128/spectrum.00656-22>

May 7, 2022

Dr. Luz Elena Vidales-Rodríguez
UNIVERSIDAD AUTONOMA DE ZACATECAS
UNIDAD ACADEMICA DE CIENCIAS BIOLÓGICAS
AVENIDA PREPARATORIA S/N COL. AGRONÓMICA II
ZACATECAS, ZACATECAS 98066
Mexico

Re: Spectrum00656-22 (*Bacillus megaterium* HgT21: a promising metal multiresistant plant growth promoting bacteria for soil bioremediation.)

Dear Dr. Luz Elena Vidales-Rodríguez:

Thank you for submitting your manuscript to Microbiology Spectrum. Three reviewers have assessed your manuscript and suggest that a revised manuscript could address all of their issues. Please carefully review the reviewer comments and prepare a revision that addresses each comment.

Link Not Available

Sincerely,

Benjamin Wolfe

Journals Department
Reviewer comments:

Reviewer #1 (Comments for the Author):

This manuscript is focused on characterization of *Bacillus megaterium* HgT21 which is known as plant growth promoting and metal tolerant bacterium. This bacterium has been widely studied because of its tolerance to different metals such as Ni, Cd, Pb, Cu and Zn. Very little is known about the tolerance mechanism of this bacterium to Hg. The authors characterized this bacterium phenotypically and

genotypically. The manuscript is well written, and the discussion is very thorough. There are some points that I thought it is worth to mention.

1) There are some details missing in "Materials and Methods" section. For example, in "Isolation of the mercury-tolerant HgT21 bacterial strain" line 108, it is not clear what are the concentrations of HgCl₂ in LB agar plates (how many plates were used and how much was the concentration in each plate). This is the same for Mercury tolerance in liquid media. What were the concentrations of HgCl₂ in liquid LB media? It is better to mention these kinds of details to make it stronger manuscript for other scientists who want to replicate the experiment.

2) Referring to the previous comment, in "Materials and Methods" section, in "Analysis of plant growth promotion by the HgT21 strain" it is unclear whether *Arabidopsis* seeds were surface sterilized before germinating on MS media or not. If the seeds were surface sterilized, what was the method? For bacterial inoculation, it is important to surface sterilize seeds to avoid other bacterial or fungal growth.

3) IAA is known as plant growth promoting hormone that stimulates root growth. In this manuscript, there is no experimental confirmation of this. It is only mentioned the genes that are involved in IAA production in *Bacillus megaterium* HgT21. It was nice if the authors showed the increase of root growth in *A. thaliana* in addition to increased wet and dry weights.

4) In Figure 6, both the figure legend and the manuscript (line 309), the statistical test that shows the wet and dry weight of the plants exposed to direct and distant interaction with HgT21 cells was increased respecting to the control is missing. It is not clear what kind of statistical test was done.

5) In the discussion, the authors did not discuss the behavior of *Bacillus megaterium* HgT21 in larger scale such as soil environment where other microorganisms are existed. In line 529, based on the characterization of this bacterium, they concluded that "*B. megaterium* HgT21 is suitable to be used as a biofertilizer in promotion of plant growth in metal contaminated environments." That would be logical if they mentioned their expectations about interactions of this bacterium with other microorganisms and its behavior when exposed to toxic metals in larger scale, such as soil environment.

Reviewer #2 (Comments for the Author):

The manuscript "*Bacillus megaterium* HgT21: a promising metal multiresistant plant growth promoting bacteria for soil bioremediation", details the genomic characteristics of a Hg resistant bacteria and its application in soil restoration.

Overall, the concept of the manuscript is good, experiment planning and execution is also good. However, the writing needs significant improvements. There are grammatical and sentence formation errors in almost every other sentence, which makes the reading very challenging.

Only doubt I have about the manuscript is, why did the authors consider the weight of the plant rather than the length of the root. Also, from figure 6, it appears that the plant roots grow until the bacterial 'straight' streak, and do not grow any further. As in the control plate, the root grows almost the entire length of the plate. Kindly explain this result in more detail.

Reviewer #3 (Comments for the Author):

This manuscript does a great job combining a whole genome sequence with various experiments (tests of metal resistance, measurement of phytohormones, plant growth promotion, etc.) to illustrate the potential of a *Bacillus* species to be useful in soil bioremediation. The writing is generally quite clear throughout the manuscript, the methods used were appropriately applied, and the data were interpreted thoughtfully. The figures are generally clear and useful.

For the lethal dose data presented in Figure 5, it would be very useful to have another species or strain for comparison to understand how this new isolate compares to a previously characterized isolate. The authors do note previously measured MICs in their discussion, but they do not have a positive control or standard used in their experiments.

It is unfortunate that the experiments to test plant growth promotion did not separate out the volatile vs. non-volatile contributions. Based on the way the experiment was set up, all of the growth promotion effects could have been due to the production of VOCs and not diffusible compounds that directly impact the plant through the agar in the plates. The authors should note this.

The plant growth promotion experiments were also very preliminary and based on responses in an agar Petri dish environment. The authors should note that more work with more plants in more realistic environments will be needed to understand the nature and extent of the plant growth promoting abilities of this bacterial isolate.

The authors should be really careful in their discussion to note correlate the presence of genes with causation of bacterial traits. They would need to conduct knockout experiments to truly demonstrate how specific genes confer resistance or other traits to the bacterium. They did not do that in this work and should therefore be careful with how they interpret their findings.

Staff Comments:

Preparing Revision Guidelines

Please return the manuscript within 60 days; if you cannot complete the modification within this time period, please contact me. If you do not wish to modify the manuscript and prefer to submit it to another journal, please notify me of your decision immediately so that the manuscript may be formally withdrawn from consideration by Microbiology Spectrum.

This manuscript is focused on characterization of *Bacillus megaterium* HgT21 which is known as plant growth promoting and metal tolerant bacterium. This bacterium has been widely studied because of its tolerance to different metals such as Ni, Cd, Pb, Cu and Zn. Very little is known about the tolerance mechanism of this bacterium to Hg. The authors characterized this bacterium phenotypically and genotypically. The manuscript is well written, and the discussion is very thorough. There are some points that I thought it is worth to mention.

1) There are some details missing in “Materials and Methods” section. For example, in “Isolation of the mercury-tolerant HgT21 bacterial strain” line 108, it is not clear what are the concentrations of HgCl₂ in LB agar plates (how many plates were used and how much was the concentration in each plate). This is the same for Mercury tolerance in liquid media. What were the concentrations of HgCl₂ in liquid LB media? It is better to mention these kinds of details to make it stronger manuscript for other scientists who want to replicate the experiment.

2) Referring to the previous comment, in “Materials and Methods” section, in “Analysis of plant growth promotion by the HgT21 strain” it is unclear whether *Arabidopsis* seeds were surface sterilized before germinating on MS media or not. If the seeds were surface sterilized, what was the method? For bacterial inoculation, it is important to surface sterilize seeds to avoid other bacterial or fungal growth.

3) IAA is known as plant growth promoting hormone that stimulates root growth. In this manuscript, there is no experimental confirmation of this. It is only mentioned the genes that are involved in IAA production in *Bacillus megaterium* HgT21. It was nice if the authors showed the increase of root growth in *A. thaliana* in addition to increased wet and dry weights.

4) In Figure 6, both the figure legend and the manuscript (line 309), the statistical test that shows the wet and dry weight of the plants exposed to direct and distant interaction with HgT21 cells was increased respecting to the control is missing. It is not clear what kind of statistical test was done.

5) In the discussion, the authors did not discuss the behavior of *Bacillus megaterium* HgT21 in larger scale such as soil environment where other microorganisms are existed. In line 529, based on the characterization of this bacterium, they concluded that “*B. megaterium* HgT21 is suitable to be used as a biofertilizer in promotion of plant growth in metal contaminated environments.” That would be logical if they mentioned their expectations about interactions of this bacterium with other microorganisms and its behavior when exposed to toxic metals in larger scale, such as soil environment.

We want to thank the reviewers for their critical and constructive comments to our manuscript. Below you will find a point-by-point response to their comments.

Reviewer #1

Comment 1.

There are some details missing in "Materials and Methods" section. For example, in "Isolation of the mercury-tolerant HgT21 bacterial strain" line 108, it is not clear what are the concentrations of HgCl₂ in LB agar plates (how many plates were used and how much was the concentration in each plate).

This is the same for Mercury tolerance in liquid media. What were the concentrations of HgCl₂ in liquid LB media? It is better to mention these kinds of details to make it stronger manuscript for other scientists who want to replicate the experiment.

R: In order to attend the reviewer's suggestion about details in methodology, data about the HgCl₂ concentrations used for MIC determination in both solid and liquid media were indicated, and the text was modified as follows:

Lines 106 to 118

"To test the HgT21 tolerance to mercury ions (Hg²⁺) in solid media, the minimum inhibitory concentration (MIC) was determined. Briefly, LB agar was supplemented with HgCl₂ at final concentrations of 0, 1, 10, 100, 200, 400, 800, 850, 900, 925, 950, 975 and 1000 μM and poured off into Petri dishes. After agar solidification, the plates were inoculated with 1X10⁵ UFC and spread onto the agar surface and incubated at 37 °C until visible growth of bacteria was detected. The concentration at which no growth was detected was reported as the MIC in solid media. Mercury tolerance in liquid media was also determined by assessing the MIC value as follows: overnight cultures were diluted 1:100 in fresh LB media supplemented with HgCl₂ at final concentrations of 0, 1, 10, 25, 50, 75 and 100 μM and grown at 37 °C and 200 rpm for 8 h (when the end of exponential growth was reached in nonexposed cultures to mercury ions), and then the optical density was spectroscopically determined at 600 nm (O.D._{600nm}). Three independent experiments were carried out."

Comment 2.

Referring to the previous comment, in "Materials and Methods" section, in "Analysis of plant growth promotion by the HgT21 strain" it is unclear whether Arabidopsis seeds were surface sterilized before germinating on MS media or not. If the seeds were surface sterilized, what was the method? For bacterial inoculation, it is important to surface sterilize seeds to avoid other bacterial or fungal growth.

R: Yes, the *A. thaliana* seeds were surface sterilized before germination. We omitted this information as the reviewer correctly pointed out. In order to improve the description of this step, we included the information about how the *A. thaliana* seeds were treated before the germination.

Lines 183 to 193

“Seeds of *Arabidopsis thaliana* ecotype Col-0 were surface-sterilized with 95% (v/v) ethanol solution for 5 minutes and bleach 20% (v/v; equivalent to 0.13 M sodium hypochlorite) for 7 minutes. Subsequently, the seeds were washed five times with sterile distilled water and stored at 4 °C for 48 h in the dark to promote and synchronize germination. The stratified *A. thaliana* seeds were placed and germinated on Petri dishes containing 0.2 X Murashige and Skoog medium (MS), pH 5.7, and 1% agar TC (micropropagation grade, PhytoTechnology Laboratories) in a horizontal arrangement on the upper zone (eight plants in each plate with 2 cm interdistance) before bacterial inoculation (46). Plates were sealed with plastic film to avoid seed contamination and placed vertically with an angle of 65° in a growth chamber (Percival Scientific) at 22-20 °C with a photoperiod of 16 h of light and 8 h darkness.”

A reference (46) was added in text and in the reference list.

López-Bucio J, Hernández-Abreu E, Sánchez-Calderón L, Nieto-Jacobo MF, Simpson J, Herrera-Estrella L (2002) Phosphate availability alters architecture and causes changes in hormone sensitivity in the *Arabidopsis* root system. *Plant Physiol* 129:244–256.

Comment 3.

IAA is known as plant growth promoting hormone that stimulates root growth. In this manuscript, there is no experimental confirmation of this. It is only mentioned the genes that are involved in IAA production in *Bacillus megaterium* HgT21. It was nice if the authors showed the increase of root growth in *A. thaliana* in addition to increased wet and dry weights.

R: More than an increase of the primary root, we observed that *B. megaterium* HgT21 induces a development of lateral roots in both contact and distant interactions, to show this, we include data of primary root length, and the number and density of lateral roots in figure 6 (panel 6D). However, please consider that a detailed analysis about the effects of *B. megaterium* over several characteristics of the radical system of the plant is under investigation.

In order to improve the description of this step in the material and methods section, we included information, specifically:

Lines 194 to 198:

“The plant growth promotion of the HgT21 strain on *A. thaliana* was carried out by in vitro direct (in contact) and indirect (distant) interactions of plant root tips with bacterial cells as previously described (47). For this analysis, 6 days old plates with *A. thaliana* seedlings were inoculated by streaking with 10 µL (approximately 1×10^6 UFC/mL) of a 24 h bacterial culture of the HgT21 strain using...”

The reference (47) added to the text was added to the reference list:

Ortiz-Castro, R., Díaz-Pérez, C., Martínez-Trujillo, M., del Río, R. E., Campos-García, J., & López-Bucio, J. (2011). Transkingdom signaling based on bacterial cyclodipeptides with auxin activity in plants. *Proceedings of the National Academy of Sciences*, 108(17), 7253-7258.

Lines 204 to 209 (addition of text)

“...with a photoperiod of 16 h of light and 8 h of darkness. Every two days, along 8 days postinoculation, photographs of the plates were taken to determine primary root length, number and density of lateral roots (density = number of lateral roots / length of primary root) using ImageJ software (available at <http://rsb.info.nih.gov/nih-image/>). In addition, seedlings were removed from the plates, weighing and dry at 80 °C in paper bags until a constant weight. The wet and the dry biomass was determined. All experiments were replicated at least three times.”

In the results, a mention of the data included in figure 6 was added:

Lines 320 to 324

“After bacterial exposure, a highly branched root system and leafy shoot were observed respecting to the control plants (Fig. 6A). Particularly, an increased number and density of lateral roots were observed in both types of interaction (contact and distant), whereas the inhibition of primary root elongation was evident when roots are in contact with bacteria (Fig. 6A, D).”

In the legend of figure 6 text was added:

Lines 939 to 940

“Primary root length, number and density of lateral root of *A. thaliana* plants after 8 days of growth in the presence of HgT21 cells (D).”

Comment 4.

In Figure 6, both the figure legend and the manuscript (line 309), the statistical test that shows the wet and dry weight of the plants exposed to direct and distant interaction with HgT21 cells was increased respecting to the control is missing. It is not clear what kind of statistical test was done.

R: As the reviewer correctly pointed out, description of the statistical analysis was missing. In order to attend this comment, we include the following information in the manuscript.

Lines 209 to 212 (addition of text)

“The differences in the number and density of lateral roots, long of primary root, and the wet and dry weights of plants exposed and not exposed (control) to direct and distant interactions with HgT21 were calculated by performing one-way analysis of variance (ANOVA) in Microsoft Excel with significance set at $P < 0.05$.”

Legend of Figure 6 (addition of text in Lines 941)

Asterisks (*) indicate significant differences determined by ANOVA ($p < 0.05$).”

Comment 5

In the discussion, the authors did not discuss the behavior of *Bacillus megaterium* HgT21 in larger scale such as soil environment where other microorganisms are existed.

In line 529, based on the characterization of this bacterium, they concluded that "B. megaterium HgT21 is suitable to be used as a biofertilizer in promotion of plant growth in metal contaminated environments." That would be logical if they mentioned their expectations about interactions of this bacterium with other microorganisms and its behavior when exposed to toxic metals in larger scale, such as soil environment.

R: We appreciate the reviewer observation. To attend this comment, we added the following information at the end of the discussion:

Line 562 to 586.

“The *Bacillus* species in the soil habitat are well recognized as “zymogenous” bacteria and ecologically have been defined as “r-strategists”, which means that they can grow quickly when the nutrient supply is abundant, as in the rhizosphere, and possess a high colonization and competitive ability. Importantly, quorum sensing-mediated processes in *Bacillus* species, such as endospore and biofilm formation, constitute an important survival strategy in soil under nutrient-limited conditions and hostile environments; its high adaptability to the environment is evidenced by its ubiquity in both nonextreme and extreme soils (109). In this context, the genetic information of this bacterium and the preliminary physiological trait analysis (metal multiresistance, plant growth promotion, IAA production, use of carbon sources) suggest a high adaptability of *B. megaterium* HgT21 to adverse environments such

as metal-contaminated soils and its potential as a PGPB through phytohormone and VOC production, phosphate solubilization and antibiotic production. Considering the essential characteristics that help to define new PGPB as biofertilizers (high rhizosphere competence, ability to increase plant biomass, long-term survival, plant-beneficial physiological traits, a lack of risk factors for human and environmental health, and high tolerance to environmental stresses encountered in soil/plant systems) (109), the genetic information of the HgT21 strain suggests that it could be suitable for use as a biofertilizer in the promotion of plant growth in metal-contaminated environments due to its ability to produce/perform quorum sensing-related processes, endospores, flagella, siderophores, metabolism of a variety of organic compounds, IAA production, and resistance to a variety of metals. However, *in vivo* experiments under more realistic conditions must be conducted to demonstrate that it can be used as a good biofertilizer. Moreover, due to its sporulating, nonpathogenic and free-endotoxin nature, *B. megaterium* HgT21 could be a good candidate for biotechnology applications in the food and pharmaceutical industries, among others (59).”

The new reference 109 in text was added to the reference list:

Van Elsas JD, Hartmann A, Schloter M, Trevors JT, Jansson JK. 2019. The Bacteria and Archaea in Soil, p. 50-54. *In* van Elsas *et. al.* (eds), Modern Soil Microbiology. CRC Press. Boca Raton, FL.

Reviewer #2 (Comments for the Author):

The manuscript "Bacillus megaterium HgT21: a promising metal multiresistant plant growth promoting bacteria for soil bioremediation", details the genomic characteristics of a Hg resistant bacteria and its application in soil restoration.

Overall, the concept of the manuscript is good, experiment planning and execution is also good. However, the writing needs significant improvements. There are grammatical and sentence formation errors in almost every other sentence, which makes the reading very challenging.

R: We appreciate this comment, to attend it we send the manuscript for English language edition to the American Journal Editors service. Certificate of the AJE edition is included.

Only doubt I have about the manuscript is, why did the authors consider the weight of the plant rather than the length of the root.

R: For this study, the plant weight was selected as an overall measure of the plant growth including both roots and shoots, however, it was observed that more than an

increase of the primary root, *B. megaterium* HgT21 induces a development of lateral roots in both contact and distant interactions, to show this, we include data of the length of primary root, number and density of lateral roots in figure 6 (figure 6D).

Due to the data added to figure 6 (Fig. 6D), some information in the manuscript was included:

Lines 204 to 210 (addition of text)

“...with a photoperiod of 16 h of light and 8 h of darkness. Every two days, along 8 days postinoculation, photographs of the plates were taken to determine primary root length, number and density of lateral roots (density = number of lateral roots / length of primary root) using ImageJ software (available at <http://rsb.info.nih.gov/nih-image/>). In addition, seedlings were removed from the plates, weighing and dry at 80 °C in paper bags until a constant weight. The wet and the dry biomass was determined. All experiments were replicated at least three times.”

Lines 320-324

“After bacterial exposure, a highly branched root system and leafy shoot were observed respecting to the control plants (Fig. 6A). Particularly, an increased number and density of lateral roots were observed in both types of interaction (contact and distant), whereas the inhibition of primary root elongation was evident when roots are in contact with bacteria (Fig. 6A, D).”

Instead of

“After bacterial exposure, significant changes in the root system architecture and shoot were observed with respect to the control plants (Fig. 6A)”. In particular, an increased number and length of lateral roots and root hairs were observed in both types of interactions (contact and distant) (Fig. 6A).

Also, from figure 6, it appears that the plant roots grow until the bacterial 'straight' streak, and do not grow any further. As in the control plate, the root grows almost the entire length of the plate. Kindly explain this result in more detail.

R: As correctly *B. megaterium* pointed out, proximity of bacterial streak inhibits further growth of the primary root. To this respect, it has been described that PGPB induce two general phenotypes on Primary Root Development in *A. thaliana*, the most common is an inhibition of primary root growth coupled with proliferation of lateral roots and root hairs leading to increased shoot biomass. The second phenotype is

an increase in plant biomass coupled to an increase in primary root growth. These phenotypes are dependent on both the bacterial density and the distance from the plant root at which the bacteria are applied (Verbon and Liberman 2016). In this study, the first phenotype was observed in *A. thaliana* after the exposition to *B. megaterium* HgT21, this result is in agreement with previous reports which describe that *B. megaterium* decreases the length of the primary root when applied directly to the roots by decreasing cell elongation in the elongation zone by 70%, whereas the root hair density increases upon colonization due to a higher number of cortical cells around the radial axis (López-Bucio et al. 2007).

Moreover, it has been described that some PGPB indirectly reduces the effect of ethylene production (an inhibitor of the elongation of primary root) through the production of the ACC deaminase which degrades ethylene and in consequence induces primary root elongation (Qin et al. 2019), however, the ACC deaminase was not found in the HgT21 genome, suggesting that the absence of the ACC deaminase during the plant-bacteria interaction could contribute to the inhibition of the root elongation.

References cited in the response to the comment:

Verbon, E. H., & Liberman, L. M. (2016). Beneficial microbes affect endogenous mechanisms controlling root development. *Trends in plant science*, 21(3), 218-2.

Qin, H., Ma, C., Zhou, Y., Miao, Y., & Huang, R. (2020). Molecular modulation of root development by ethylene. *Small Methods*, 4(8), 1900067.

To attend this comment, the following text was added to the manuscript:

Lines 412-420

“The phenotype observed in *A. thaliana* after the interaction with *B. megaterium* HgT21 correlates with previous reports that describe the increase of lateral roots, decreasing cell elongation in primary root, and root hair density increase upon colonization (64). In relation to the phenotype observed, it has been described that some PGPB indirectly reduces the effect of ethylene production (an inhibitor of primary growth root) through the production of the ACC deaminase which degrades ethylene, and in consequence induces primary root elongation (65). However, the gene encoding for the ACC deaminase was not found in *B.*

megaterium HgT21, this could explain the inhibition of the root growth when colonization is established.”

The references (64-65) added to the text were added to the reference list:

López-Bucio, J., Campos-Cuevas, J. C., Hernández-Calderón, E., Velásquez-Becerra, C., Farías-Rodríguez, R., Macías-Rodríguez, L. I., & Valencia-Cantero, E. (2007). *Bacillus megaterium* rhizobacteria promote growth and alter root-system architecture through an auxin-and ethylene-independent signaling mechanism in *Arabidopsis thaliana*. *Molecular Plant-Microbe Interactions*, 20(2), 207-217.

Qin, H., Ma, C., Zhou, Y., Miao, Y., & Huang, R. (2020). Molecular modulation of root development by ethylene. *Small Methods*, 4(8), 1900067.

Reviewer #3 (Comments for the Author):

This manuscript does a great job combining a whole genome sequence with various experiments (tests of metal resistance, measurement of phytohormones, plant growth promotion, etc.) to illustrate the potential of a *Bacillus* species to be useful in soil bioremediation. The writing is generally quite clear throughout the manuscript, the methods used were appropriately applied, and the data were interpreted thoughtfully. The figures are generally clear and useful.

For the lethal dose data presented in Figure 5, it would be very useful to have another species or strain for comparison to understand how this new isolate compares to a previously characterized isolate. The authors do note previously measured MICs in their discussion, but they do not have a positive control or standard used in their experiments.

R: We want to thank for this observation to the reviewer, we had not even considered include a metal resistant type strain for comparison. As correctly pointed out, no positive or standard strain was used as control to compare the MICs, instead of, levels of resistance were compared with those reported in different metal resistant *Bacillus* strains. This study was conceptualized on this way due to at least to our knowledge, no metal multiresistance gram-positive strains has been reported or available in Type Culture Collections, however, we agree that it would be useful and interesting to compare the MICs of HgT21 with the MICs obtained experimentally under the same conditions of a metal resistant prototype (i.e. the multiresistant Gramnegative strain *Cupriavidus metallidurans* (Goris et al.) Vandamme and Coenye (ATCC 43123), and/or with resistant strains for each of the metal tested in this study. Definitely, acquisition of type metal resistant strains will be necessary in future work, especially in studies related to the ability of this strain to cope with environmental toxic concentrations of heavy metals.

It is unfortunate that the experiments to test plant growth promotion did not separate out the volatile vs. non-volatile contributions. Based on the way the experiment was set up, all of the growth promotion effects could have been due to the production of VOCs and not

diffusible compounds that directly impact the plant through the agar in the plates. The authors should note this.

R: We agree with this comment, however, characterization of the mechanisms by which the HgT21 strain promotes growth of plants will be done in future work. Specifically, to elucidate the contribution of VOCs and diffusible compounds to plant growth, *in vitro* experiments using divided Petri dishes to avoid diffusion of compounds across the agar media will be necessary, as well as the direct determination of VOCs during the interaction plant/bacteria.

In order to wide this point in the manuscript, text was modified as follows:

Lines 424-426

...“however, this must be experimentally demonstrated and future experiments must be conducted to evaluate the contribution of VOCs and diffusible compounds to growth promotion.”

instead of:

...“however, this must be experimentally demonstrated.”

The plant growth promotion experiments were also very preliminary and based on responses in agar Petri dish environment. The authors should note that more work with more plants in more realistic environments will be needed to understand the nature and extent of the plant growth promoting abilities of this bacterial isolate.

R: Yes, we absolutely agree with this comment, a wide of characteristics must be taken into account to consider a bacterial strain as a good biofertilizer, (high rhizosphere competence, ability to increase plant biomass, long term survival, plant-beneficial physiological traits, a lack of risk factors for human and environmental health, and high tolerance to environmental stresses encountered in soil/plant systems), we are conscious that plant growth promotion by this strain must be demonstrated *in vivo* under field conditions in a more complex environment that in this case could include the presence of toxic metals. Importantly, mechanisms involved in plant growth promotion under real conditions must be elucidated in future.

We consider that results presented in this work strongly suggest that HgT21 strain possess important genetic and physiological traits that could be important to face the adverse conditions that prevails in metal contaminated soils and could contribute synergistically to alleviate the stress in plants caused by the presence of toxic metals and to promote the plant growth. However, in order to point out the importance of the study of *in vivo* experiments under realistic conditions, in the discussion section of the manuscript, text was modified as follows:

Lines 574 to 583

“Considering the essential characteristics that help to define new PGPB as biofertilizers (high rhizosphere competence, ability to increase plant biomass, long-term survival, plant-

beneficial physiological traits, a lack of risk factors for human and environmental health, and high tolerance to environmental stresses encountered in soil/plant systems) (109), the genetic information of the HgT21 strain suggests that it could be suitable for use as a biofertilizer in the promotion of plant growth in metal-contaminated environments due to its ability to produce/perform quorum sensing-related processes, endospores, flagella, siderophores, metabolism of a variety of organic compounds, IAA production, and resistance to a variety of metals. However, *in vivo* experiments under more realistic conditions must be conducted to demonstrate that it can be used as a good biofertilizer.”

The new reference 109 in text was added to the reference list:

Van Elsas JD, Hartmann A, Schloter M, Trevors JT, Jansson JK. 2019. The Bacteria and Archaea in Soil, p. 50-54. *In* van Elsas *et. al.* (eds), Modern Soil Microbiology. CRC Press. Boca Raton, FL.

The authors should be really careful in their discussion to note correlate the presence of genes with causation of bacterial traits. They would need to conduct knockout experiments to truly demonstrate how specific genes confer resistance or other traits to the bacterium. They did not do that in this work and should therefore be careful with how they interpret their findings.

R: In order to attend these observations, the discussion was modified in several points as follows:

Lines 403 to 404

“These results suggest that enzymes encoded by genes of the chorismate pathway could be involved in IAA production by *B. megaterium* HgT21.”

instead of

“In agreement with these results, the production of IAA by *B. megaterium* HgT21 was demonstrated.”

Lines 409-410

The following text was added:

“Although the activity of these enzymes was not demonstrated in this work, the presence of their encoding genes in the HgT21 genome, could be related to....”

Lines 465 to 467

“The enzymes encoded by the arsenate resistance like-operons in HgT21 could be responsible for the observed arsenate resistance in this strain, which is similar to the most arsenic-resistant *Bacillus* species”

instead of

“...and explain the observed resistance to arsenate, which is similar to the most arsenic-resistant *Bacillus* species”.

Line 524

The following text was added:

“...however, this must be experimentally demonstrated.”

Lines 549 to 551

The following text was added:

“However, the function of enzymes encoded in tellurite resistance-related genes in *B. megaterium* HgT21 must be experimentally demonstrated.”

July 26, 2022

Dr. Luz Elena Vidales-Rodríguez
UNIVERSIDAD AUTONOMA DE ZACATECAS
UNIDAD ACADEMICA DE CIENCIAS BIOLÓGICAS
AVENIDA PREPARATORIA S/N COL. AGRONÓMICA II
ZACATECAS, ZACATECAS 98066
Mexico

Re: Spectrum00656-22R1 (*Bacillus megaterium* HgT21: A promising metal multiresistant plant growth promoting bacteria for soil bioremediation)

Dear Dr. Luz Elena Vidales-Rodríguez:

Your manuscript has been accepted, and I am forwarding it to the ASM Journals Department for publication. You will be notified when your proofs are ready to be viewed.

Sincerely,

Benjamin Wolfe
Editor, Microbiology Spectrum

Journals Department
Supplemental Material File: Accept